# Projecting long-term excess risks of major infectious diseases associated with future extreme weather events in Thailand

Esther Li Wen Choo[1], Pei Ma[2], Jo Yi Chow[1], Steve Hung-Lam Yim[1,3,4], Oliver Brady[5,6], Borame Lee Dickens[2], Jue Tao Lim[1]*

**1** Lee Kong Chian School of Medicine, Nanyang Technological University, Singapore, Singapore, **2** Saw Swee Hock School of Public Health, National University of Singapore, Singapore, Singapore, **3** Asian School of the Environment, Nanyang Technological University, Singapore, Singapore, **4** Centre for Climate Change and Environmental Health, Nanyang Technological University, Singapore, Singapore, **5** Centre for Mathematical Modelling of Infectious Diseases, London School of Hygiene and Tropical Medicine, London, United Kingdom, **6** Department of Infectious Disease Epidemiology and Dynamics, Faculty of Epidemiology and Population Health, London School of Hygiene & Tropical Medicine, London, United Kingdom

\* juetao.lim@ntu.edu.sg

## Abstract

Climate change is postulated to impact infectious disease transmission, yet few studies have characterised the excess risks of infectious diseases associated with extreme weather events. To address this, we conducted a study estimating and projecting the impacts of extreme heat and precipitation on the incidence of major infectious diseases in Thailand. We developed, fitted and validated an analytical framework to model province-level disease cases and their relationship with extreme weather indicators based on historical data. We used generalised additive models to delineate the relationship between monthly extreme heat days, standardised pre-cipitation index and incidence rates of seven infectious diseases (dengue, malaria, Japanese encephalitis, melioidosis, leptospirosis, pneumonia, influenza) across Thailand's provinces. Disease-specific models were fitted to historical surveillance data and used to project future disease incidence across 4 Shared Socioeconomic Pathways (SSP) based on MIROC6 climate projections. Historically, extreme heat was associated with an increase in all infectious disease incidences except malaria and leptospirosis. We projected that dengue risk declines in most future climate change scenarios, except SSP245 where extreme heat drives a significant rise in North-ern and Central Thailand from 2021–2060. Nationwide dengue risk is expected to decrease by 24.9% (95%CI:9.68%,40.0%) during future periods of extreme weather from 2061–2080 compared to historical baselines. Influenced by heat and dry weather in Northeastern and Central regions, influenza risk is expected to increase under SSP245 in 2021–2060, then decrease with extreme precipitation. Influenza risk in Nakhon Ratchasima is expected to increase by 36.8% (95%CI:9.83%,63.8%)

**Data availability statement:** The code and data used for this study can be accessed from https://github.com/EstherChoo/THextremeweather. Processed data can be found at http://dx.doi.org/10.6084/m9.figshare.30579395.

**Funding:** This research is funded and supported by the Lee Kong Chian School of Medicine – Ministry of Education Start-Up Grant (LJT). This research / project is also supported by the Ministry of Education, Singapore, under its Academic Research Fund Tier 1 (RT4/22) and Academic Research Fund Tier 1 Seed Funding Grant (RS04/22) (LJT). This research/project is supported by the National Research Foundation, Singapore and National Environment Agency, Singapore under the Climate Impact Science Research Funding Initiative (Award No. CISR-2023-1R-01) (LJT). OJB was supported by a UK Medical Research Council Career Development Award (MR/V031112/1). The funders had no role in study design, data collection and analysis, decision to publish, or preparation of the manuscript.

**Competing interests:** The authors have declared that no competing interests exist.

in 2021–2040 under SSP245. Localised public health interventions are necessary to address climate change impacts.

## Author summary

Climate change is increasingly recognised as having a significant influence on infectious disease spread, yet our understanding of how extreme weather conditions specifically affect disease risks remains limited. This study investigated how extreme heat and rainfall affect the incidence of nine major respiratory, food-borne, and vector-borne diseases across 77 provinces in Thailand. Using historical surveillance and climate data, we developed statistical models to describe how disease incidence responds to prolonged periods of high temperatures and fluctuations in rainfall.

We found that extended extreme heat typically led to inclines in most infectious diseases, including influenza, which increased in hot and dry conditions. Looking ahead, our models project that dengue incidence will generally fall under most future climate scenarios, with the exception of moderate warming, where cases are expected to notably increase in Northern and Central Thailand through to 2060. By the late 21st century, dengue risk during extreme weather events is projected to decline substantially compared to historical levels. In contrast, influenza is likely to rise initially with dry spells before decreasing with heavier rainfall. As the frequency and intensity of climate extremes escalate, our findings highlight the need to integrate climate resilience into local public health responses before risks further intensify.

## Introduction

Due to climate change, extreme weather events, such as heatwaves and droughts are expected to rise in frequency, duration and intensity [1], and consequentially impact the risk and burdens of infectious diseases. Historically, heatwaves have been shown to increase the risk of dengue [2], influenza [3] and malaria transmission [4]. Flooding, on the other hand, can contaminate water with harmful bacteria and viruses, increasing the transmissibility of waterborne and foodborne diseases [5]. A systematic review and meta-analysis including 106 studies demonstrated that 1 °C increase in high temperature was associated with a 13% increase in dengue risk [6]. Another study found that beyond 31 °C, a 1 °C increase in maximum temperature was associated with 13.1% decrease in cumulative dengue risk over 6 weeks [7]. Abnormal temperature fluctuations driven by climate change have also been linked to increased influenza outbreaks [7]. Additionally, a global analysis connected anomalously dry and wet weather to increased risk of infectious diseases symptoms like cough, fever and diarrhoea [8]. Flood events frequently trigger disease outbreaks as well; for example, the Pakistan July 2010 floods were followed by increased acute respiratory infection, acute diarrhoea, skin diseases, and suspected malaria [9].

Most studies have focused on the effects of mean temperature and rainfall on future disease burden. Recent work has projected the burden of influenza [10], leptospirosis [11], malaria [12] and dengue [13] based on temperature and precipitation projections under different climate change scenarios. Colón-González et al. projected the future burden of dengue in Southeast Asia based on climate, population, human mobility and gross domestic product projections – expecting it to peak in this century before declining to lower levels. Heterogenous impacts were anticipated across the region – with long term average decreases in dengue burden in Thailand and Cambodia [13]. However, while extreme weather events are expected to increase in occurrence, no study has attempted to quantify the impact of extreme weather events on climate-sensitive infectious diseases.

To address these gaps, we sought to understand how extreme weather events have influenced the historical burden of major climate-sensitive infectious diseases, as well as project the prospective burdens of these infectious diseases in periods of extreme weather in future climate change scenarios. Leveraging the broad and long temporal frame of the national disease surveillance database in Thailand, we ascertained and characterized how extreme weather events can influence the historical and future burdens and risk of a large battery of vector-, animal-, air- and food-borne climate-sensitive diseases. Thailand was an ideal focus for this study due to its high-quality, long-term data on various climate-sensitive infectious diseases, supported by well-established disease surveillance systems [14]. The country's diverse climatic zones, ranging from tropical monsoonal regions in the south to temperate highlands in the north, enabled us to study the effects of different climate exposures on disease dynamics [15]. Additionally, Thailand's active engagement in public health and climate adaptation planning provides a valuable context for integrating research findings into actionable policies [16]. We developed, calibrated and validated an analytical framework to model disease incidences and extreme weather measurements, such as extreme heat days and flood/drought scenarios based on historical data from 2003 to 2019. Based on Coupled Model Intercomparison Project Phase 6 (CMIP6) climate change projections, we used the calibrated model to predict the future excess risk of studied disease attributable to extreme weather events from 2021–2100 under 4 separate climate change scenarios.

## Methods

### Disease case data

Disease surveillance data between 2003 and 2019 was obtained from Thailand's Ministry of Public Health disease surveillance system which records reported monthly disease case counts at the province level [17]. Bueng Khan province was separated from Nong Khai in 2011, therefore for consistency over the timeframe of the dataset, we merged disease case counts from Bueng Khan back to Nong Khai from 2011 onwards. We considered seven major infectious diseases circulating in Thailand, which include the common infectious diseases that are reported by Thailand's Ministry of Public Health [17]. The diseases include Japanese encephalitis virus (JEV), malaria, dengue, pneumonia, influenza, leptospirosis and melioidosis. These diseases were chosen among the common infectious diseases in Thailand as the risk of transmission from these diseases are plausibly influenced by weather conditions [2–4,18,19]. There were no documented changes in case definitions for the diseases of interest between 2003 and 2019, and cases were reported based on clinical suspicion. We also chose a group of diseases which varied in transmission mode to investigate how the burdens of these diseases may be impacted by current and future extreme weather patterns.

### Historical meteorological data

Meteorological data from 2003 to 2019 was obtained from ERA5, published by the European Centre for Medium-Range Weather Forecasts. ERA5 provided hourly estimates across a 30km grid, which we spatially averaged for each province in Thailand [20]. Air temperature at 2m, dewpoint temperature, total precipitation was collected. The hourly estimates were used to calculate daily minimum and maximum temperature and daily total precipitation.

## Demographic data

Population size for each province from 2003 to 2019 was obtained from the Official Statistics Registration Systems of Thailand [21]. Similarly, as Bueng Khan was split from Nong Khai in 2011 to form a separate province, we merged the population from Bueng Khan back to Nong Khai from 2011 onwards, to allow for consistent analysis over the timeframe of the dataset. Annual population projections at 30-arc-seconds resolution under 4 climate change scenarios SSP126, SSP245, SSP370 and SSP585 from 2021 to 2100 were obtained, which were simulated based on the CMIP6 experimental design data and historical population from 2010 [22]. The projections were spatially aggregated to each population in Thailand. The annual projections were aggregated in 20-year periods to be consistent with the meteorological data projections. We obtained a second historical population dataset from WorldPop for each province from 2003 to 2019, which was the data used to train the CMIP6 population projection data [23]. This dataset was used exclusively to train the models in the sensitivity analysis that incorporated future population changes. For the main analysis, we used government census-based population data, which are more reliable for historical estimates but do not extend to future projections.

## Extreme weather events

Extreme weather events were classified into extreme heat, extreme dry weather and extreme wet weather. An extreme heat day was defined as a daily maximum temperature above a threshold for at least 3 consecutive days. The threshold was set at the 90th percentile of the national daily maximum temperature (34.9°C), with the reference period being 2003–2021. A single national threshold, rather than a region-specific threshold, was used to define extreme heat days to ensure comparability across regions. Furthermore, the 90th percentile daily maximum temperature differed by only 1–2 °C across geographical regions, supporting the use of a national threshold. Daily precipitation was converted to monthly Standardised Precipitation Index-1 (SPI) values; which is commonly used as a drought index and can reflect precipitation anomalies [24]. Conversions used the R package SPEI, using 2003–2021 as the reference period [25]. We used SPI rather than Standardized Precipitation-Evapotranspiration Index (SPEI) due to the limited availability of projected potential evapotranspiration under future climate change scenarios. To test if SPEI and SPI categorised wet and dry months similarly, potential evapotranspiration was estimated for the historical period by using the Hargreaves equation [26] to calculate SPEI. SPEI-1 and SPI-1 categorised extreme dry, extreme wet and normal months identically in 81% of the data. Hence, the use of SPI rather than SPEI would not have significantly affected our results. An accumulation period of 1 month was chosen to account for short-term impacts. Months with SPI lower than -1.5 were characterised as extreme dry weather and months with SPI higher than 1.5 were characterised as extreme wet weather [27].

## Meteorological data under climate change scenarios

Climate change projections were obtained from 3 general circulation models, MIROC6, CMCC-ESM2 and IPSL-CM6A-LR under 4 climate change scenarios, SSP126, SSP245, SSP370 and SSP585 [28–30]. We chose only 3 models as they covered a range of climate sensitivities and focus on different processes [28–30]. MIROC6, CMCC-ESM2 and IPSL-CM6A-LR represents low to moderate, moderate and high climate sensitivities respectively [24–26]. Projected monthly minimum temperature, maximum temperature and total precipitation were obtained across the periods of 2021–2040, 2041–2060, 2061–2080 and 2081–2100, at a 30-arc-second resolution. This means that the dataset provided the average projection of each calendar month over 20 years. MODAWEC, a daily weather generator, was used to disaggregate daily precipitation, daily maximum and daily minimum temperature from the respective monthly data [31]. The monthly number of wet days was also required as an input by MODAWEC, where a wet day is defined as a day with more than 1 mm of precipitation, which is consistent with standard definitions and climate studies. The future number of wet days were linearly interpolated based on historical daily precipitation data and number of wet days. Only the daily maximum temperature was used in subsequent analyses to determine the monthly number of extreme heat days.

## Climate change scenarios

The Shared Socioeconomic Pathway (SSP) framework developed for the CMIP6 provides potential future scenarios based on socioeconomic factors and greenhouse gas concentrations [1]. In this study, the SSP126, SSP245, SSP370 and SSP585 scenarios were considered, which are summarised in Table 1 [32].

## Statistical model

Generalised additive models (GAMs) with a negative binomial distribution were used to model the associations between disease case counts and the covariates. The covariates include lagged monthly SPI, lagged number of monthly extreme heat days, population density and contemporaneous relative humidity. The negative binomial link function was chosen as it was the appropriate distribution to model the case counts which were overdispersed based on the Cameron and Trivedi test [33] (S1 Table). The models were trained on data from 2003 to 2019. GAM has been employed in other similar studies to project future disease incidence using meteorological variables and to study the association between disease incidence and these factors [34–36]. GAMs offer the advantage of flexibility by modelling predictors as smooth non-linear functions, as opposed to only linearly in generalised linear models (GLMs) [37]. This is particularly important given the evidence of non-linear associations between extreme weather and disease risk [35,36,38]. Spatial random effects were used to account for differences in transmission dynamics across provinces. For each province and disease, the following GAM specification was used:

$$\log (y_{t,\,d,l}) = \alpha_d + \log \left( P_{a[t],l} \right) + + \sum_{k=1}^{p} f_{1,k,d} \left( x_{1,t-k} \right) + \sum_{j=1}^{q} f_{2,j,d} \left( x_{2,t-j} \right) + f_{3,d} \left( m[t] \right) + f_{4,d} \left( x_{3,a[t],\,l} \right) + f_{5,d}(z_{t,l}) + \gamma_l$$

(1)

where $y_{t,d,l}$ is the case count of disease $d$ at monthly time $t$ in province $l$, $\alpha$ is the intercept, $\log \left( P_{a[t]} \right)$ is the logarithm of the province population at year $a[t]$ which is used as an offset for the at-risk population. $x_{1,t-k}$ is the SPI up to $k = \{0,\ 1, 2, .., p\}$ lags, $x_{2,t-j}$ is the number of extreme heat days in a month up to $j = \{0,\ 1,\ 2, ..., q\}$ lags, $x_{3,a[t],\,l}$ is the population density at year $a[t]$ in province $l$, $z_{t,l}$ is the contemporaneous relative humidity in province $z_{t,l}$, and $f_{1,k,d}$, $f_{2,j,d}$, $f_{3,d}$, $f_{4,d}$ and $f_{5,d}$ are the thin-plate splines for disease $d$. Seasonal trends were accounted for using thin-plate splines $f_{3,d}$ on each

**Table 1. Description of SSP scenarios.**

| Scenario | Radiative forcing by 2100 (W/m²) | Description |
|---|---|---|
| SSP126 | 2.6 | Low challenges to mitigation (resource efficiency) and adaptation (rapid development). A sustainable pathway where human well-being is prioritised over economic growth and climate protection measures are taken. There is a reduction in income inequality. Consumption is aimed at reducing material resource and energy usage. |
| SSP245 | 4.5 | Moderate challenges to mitigation and adaptation. "Middle of the road" pathway that extends historical and current global development. Income patterns vary greatly across countries and states cooperate only to some extent. There is an increase in global population which plateaus in the second half of the century. Environmental systems are deteriorating to some extent, but climate protection measures are taken. |
| SSP370 | 7.0 | High challenges to both mitigation and adaptation. The income and education gap between developed societies and less-developed societies are widening. Environmental policies are able to tackle local problems, but only in some regions. |
| SSP585 | 8.5 | High challenges to mitigation and low challenges to adaptation. There is a high rate of innovation and technological process. Social and economic development are dependent on an increased exploitation of fossil fuel resources, comprising of a high percentage of coal. The global economy is growing and local environmental issues like air pollution are being effectively addressed. |

calendar month $m[t]$ for disease $d$. Although the splines for calendar month were not explicitly specified as cyclic, the fitted monthly smooths naturally exhibited a cyclical pattern, appropriately capturing seasonality. The covariate lags $p$ and $q$ were chosen by the mean Akaike information criterion (AIC) across all models of a disease (S2 Table). Thin-plate splines were chosen as they produce a lower mean squared error than alternative splines [39]. A basis dimension of 10 was used for each spline which allowed for a balance of flexibility and parsimony. Spatial dependencies across provinces were accounted for using random effects $\gamma_l$ for each province $l$. Moran's I was calculated across diseases and timepoints. As the observed spatial autocorrelation was very weak and largely statistically non-significant, we did not incorporate spatial autocorrelation in the model. Restricted maximum likelihood (REML) was used to estimate the splines rather than generalised cross-validation as the latter tends to favour more complex models and under-smooth terms [40].

For dengue and JEV, lags of up to 2–3 months for extreme heat and SPI reflect their effects on the extrinsic incubation period and vector reproduction, together with the delayed effects on vector abundance [41,42]. For malaria, the short heat lag of 1 month aligns with the effect of extreme heat on the extrinsic incubation period, while rainfall effects on vector habitats accumulate over a longer period of 3 months [43,44]. For pneumonia, leptospirosis and influenza, 3-month lags for both extreme heat and SPI capture delayed effects of temperature and moisture on transmission, exposure pathways and host susceptibility [45,46]. For pneumonia and influenza, the longer lags compared to the incubation periods may be the cumulative effects of extreme weather on respiratory immunity [45,47]. Melioidosis included an immediate heat effect and a 3-month SPI lag to capture delayed rainfall impacts on soil and bacterial exposure [18].

## Estimation of incidence rate ratio and population attributable fraction

The historical effect of SPI and extreme heat days on diseases case counts were measured by the incidence rate ratio (IRR) and population attributable fraction (PAF). The IRR gives the ratio of predicted cases with exposure to predicted cases with no exposure. Specifically, the IRR for SPI gives the ratio between incidence rates at varying values of SPI and the incidence rate at 0 SPI. An SPI value of 0 represents average precipitation conditions based on the long-term mean precipitation in that location. The IRR for extreme heat gives the ratio between incidence rates at non-zero days of extreme heat in a month and the incidence rate with no days of extreme heat. In computation of IRRs, we only varied one covariate (SPI or extreme heat days) at some level $u$ while holding all other covariates at their historically observed mean value. The IRR was calculated as follows for each disease:

$$IRR_{d,l,i} = \frac{E\left[\hat{y}_{d,l}\middle| x_i = u, \ \ x_{-i} = \overline{x_{l,-i}}\right]}{E\left[\hat{y}_{d,l}\middle| x_i = 0, \ \ x_{-i} = \overline{x_{l,-i}}\right]}$$

(2)

where $\hat{y}$ is the estimated incidence rate of the disease $d$ in province $l$, $x_i$ is the covariate of interest and $x_{-i}$ are the remaining set of covariates set at the province-specific means $\overline{x_{l,-i}}$.

In the case of computing national-level IRRs, province-level incidence rates and covariates in (2) are replaced by national-level incidence rates and covariates respectively.

The PAF gives the proportion of cases in a population that is attributed to a particular risk factor. A PAF of 0 indicates that the risk factor has no impact on the occurrence of the disease, and a negative PAF indicates that the factor has a protective effect against the disease. The province-specific PAF was estimated with the ratio of predicted cases when varying either SPI or extreme heat days from zero, with other covariates held constant, against predicted cases when SPI and extreme heat days are set to zero. The province-specific PAF was calculated as follows for each disease:

$$PAF_{d,l,i} = \frac{E\left[\hat{y}_{d,l}\middle| x_i = u, \ \ x_{-i} = \overline{x_{l,-i}}\right] \ - \ E\left[\hat{y}_{d,l}\middle| x_i = 0, \ \ x_{-i} = \overline{x_{l,-i}}\right]}{E\left[\hat{y}_{d,l}\middle| x_i = 0, \ \ x_{-i} = \overline{x_{l,-i}}\right]}$$

(3)

where $\hat{y}$ is the estimated incidence rate of the disease $d$ in province $l$, $x_i$ is the covariate of interest and $x_{-i}$ are the remaining set of covariates set at the province-specific means $\overline{x_{l,-i}}$. Similar to IRR, national-level PAF is calculated by replacing province-level incidence rates and covariates in (3) by national-level incidence rates and covariates respectively. 95% confidence intervals were calculated for both IRR and PAF using the delta method, which derives the asymptotic distribution of the IRR and PAF assuming incidence rates are asymptotically Gaussian.

**Future projection and estimation of excess risk due to extreme weather scenarios**

We calibrated the statistical models to province and disease specific data from 2003–2019. Thereafter, future monthly disease cases for each province, time period and climate change scenario were projected, based on the MIROC6 climate projections of total precipitation, relative humidity and maximum temperature. We chose MIROC6 as the main CMIP6 model as it was specifically designed to capture complex Asian monsoon systems and ocean-atmosphere interactions, which are crucial for modelling Thailand's climate. The time periods considered were 2021–2040, 2041–2060, 2061–2080, 2081–2100 and the climate change scenarios considered were SSP126, SSP245, SSP370, SSP585. Projections were compared to a reference period of 2003–2019. Province population counts were set at the historical mean values. Population was assumed to remain at the historical level so that the excess risk could be solely attributed to extreme weather events.

To minimise noise, excess risk was estimated only based on timepoints where the respective extreme weather events were projected to take place. The main extreme weather exposures considered were (1) extreme heat, extreme dry and extreme wet weather, (2) extreme heat, (3) extreme dry weather and (4) extreme wet weather. Additionally, we considered sub-scenarios to verify that the impact of one extreme weather event was not confounded by another. For example, given that extreme heat is associated with extreme dry weather. the excess risk attributed to extreme heat may be due to extreme dry weather instead [48]. Hence, we considered the following sub-scenarios – (1) extreme heat and no extreme dry weather, (2) extreme heat and no extreme wet weather, (3) extreme heat with no extreme dry and no extreme wet weather, (4) extreme wet weather and no extreme heat, and (5) extreme dry weather and no extreme heat. The presence of extreme heat refers to one or more extreme heat days in a month. We also estimated the projected disease case counts and excess risk based on 2 alternative climate change models, CMCC-ESM2 and IPSL-CM6A-LR, for a sensitivity analysis. Using the disease-specific GAM models, the projected province-level incidence of each disease at each future time period and climate change scenario, and the corresponding fitted historical incidence is estimated as follows:

$$\hat{y}_{proj,d,m,t,l,s} = E\left[\hat{y}_{d,m,t,l,s}\big|x_i = x^*_{i,m,t,l,s}, \left\{x^*_{1,m-k,t,l,s}\big|1 \le k \le p\right\}, \left\{x^*_{2,m-j,t,l,s}\big|1 \le j \le q\right\}, x_{-i} = \overline{x_{-i,m,l}}\right] \tag{4}$$

$$y_{hist,d,m,l} = E\left[\hat{y}_{d,m,l}\big|x = \overline{x_{m,l}}\right] \tag{5}$$

where $\hat{y}$ is the estimated incidence rate of the disease $d$ in month $m$ of future time period $t$ and province $l$ under climate change scenario $s$. $x_i$ include the covariates SPI, number of extreme heat days and relative humidity, and $x^*_{i,s,m}$ is the covariate of interest $i$ in month $m$ of future time period $t$ under climate change scenario $s$ in province $l$. $x^*_{1,m-k,t,l,s}$ is the lagged monthly extreme heat days of up to $p$ lags, while $x^*_{2,m-q,t,l,s}$ is the lagged monthly extreme heat days of up to $q$ lags. The number of lags are shown in S2 Table. As climate projections only provide multi-year monthly averages rather than the continuous monthly time series, lagged covariates are treated as cyclical. For example, for disease projections in February, 3 months of lagged SPI would be taken from SPI in January, December and November. $x_{-i}$ are the remaining covariates and $\overline{x_{m,l}}$ are the mean of the covariates in month $m$ over the historical time period 2003–2019 in province $l$. The fitted historical incidence is estimated based on $\overline{x_{m,l}}$, the mean of all covariates in month $m$ of the historical time period 2003–2019 in province $l$. 95% prediction intervals were calculated for the excess risk estimates by using the 95% prediction intervals of the projected disease case counts.

Annual province-level ER attributable to extreme weather is calculated using the incidences obtained from (4) and (5), aggregated over months where extreme weather is projected to occur. For each month, the projected incidence under future weather conditions was compared with the expected incidence under historical average conditions for that same month. This ensures that excess risk reflects the impact of future extreme weather while holding all other covariates constant.

$$\text{Province } ER_{d,t,l,s,q} = \frac{\sum_{m \in M_{q,t,s,l}} y_{proj,d,m,t,l,s} - \sum_{m \in M_{q,t,s,l}} y_{hist,d,m,l}}{\sum_{m \in M_{q,t,s,l}} y_{hist,d,m,l}} \times 100\%$$

(6)

$$q \in \{(SPI \leq -1.5 \cup SPI \geq 1.5 \cup ExtremeHeatDays > 0), \ SPI \leq -1.5, \ SPI \geq 1.5, \ ExtremeHeatDays > 0\}$$

(7)

where $M_q$ are the months where extreme weather condition $q$ occurs in future time period $t$ under climate change scenario $s$ in province $l$, which includes months with all extreme weather events, extreme dry weather, extreme wet weather and extreme heat respectively. The extreme weather conditions are explicitly defined in (7). This restricts our estimation of ER to include only incidence in months which extreme weather takes place.

Annual national-level ER is calculated by modifying (6) such that incidences are aggregated over all provinces. This represents the percentage change in projected nationwide incidence under extreme weather conditions compared to the expected nationwide incidence under historical average weather.

## Model evaluation

For each disease, the Akaike information criterion (AIC) of models with different variable lags were compared, where the model with the lowest AIC was selected for the rest of the analysis. Model fit was evaluated using AIC as it balances model fit with complexity and is more suitable for long-term projections than time series cross-validation approaches, which are intended for short-term forecasts. The in-sample fit of the models were assessed by AIC and root mean squared error (RMSE). For each disease, the AIC and RMSE of the GAM model were compared against corresponding generalised linear models (GLMs) with the same covariates and lags, as chosen by the optimal AIC previously, without the smooth function.

As part of the sensitivity analysis, we also developed models which included future population as an offset and future population density as a predictor instead of historical population. We compared the projections of the models with future population against those using historical population for each disease. Additionally, we trained the models using three different sets of global climate model (GCM) projections, MIROC6, CMCC-ESM2 and IPSL-CM6A-LR, and compared the national-level excess risks attributable to extreme weather derived from each dataset across time and climate change scenarios. This ensured that our results were not heavily influenced by the choice of GCM used for our main results, which was MIROC6. Furthermore, we utilised the delta method to compute the confidence intervals of our IRR and PAF estimates, allowing us to appropriately characterise the uncertainty of the association between extreme weather and disease incidence rate.

## Results

### Future climate change scenarios

In this study, we considered 4 of the CMIP6 climate change scenarios - SSP126, SSP245, SSP370, SSP585, which take into account a wide range of socioeconomic trends and radiative forcings (Table 1). Temperatures are expected to increase across SSPs, where SSP585 has the largest increase in extreme heat days from 2021 to 2100 across all provinces (Fig 1). A similar trend is observed on a provincial scale (S1 Fig). The national level precipitation remains largely the

same across SSPs over time, with similar minimum and maximum monthly SPI values among all the SSPs (Fig 1). From 2021 to 2100, extreme SPI values are only observed at the province level (S1 Fig).

## Association of diseases with extreme weather

We developed generalised additive models for province-level disease cases and trained them on extreme weather indicators from 2003–2021 to obtain the relationship between extreme weather and disease case counts. We expressed the relationship between extreme weather and disease incidence in incidence rate ratios (IRR) and population attributable fractions (PAF). IRR can be interpreted as the ratio change in incidence rates given exposure to extreme weather in the same month compared to the baseline (0 heat days and 0 SPI respectively), while PAF can be interpreted as the proportion of cases in a population that could be avoided in the absence of anomalous SPI and extreme heat in the same month.

In dengue, JEV, influenza, pneumonia and melioidosis, the number of extreme heat days in a month was positively associated with disease risk (Fig 2). On the other hand, increasing days of extreme heat days in a month was negatively associated with malaria and leptospirosis risk (Fig 2). The pattern was also consistent when the association of disease incidence against extreme weather was broken down on a province-specific basis, particularly in highly populated provinces like Bangkok, Nakhon Ratchasima and Ubon Ratchathani (S2–S10 Figs). Province-level PAFs demonstrated the

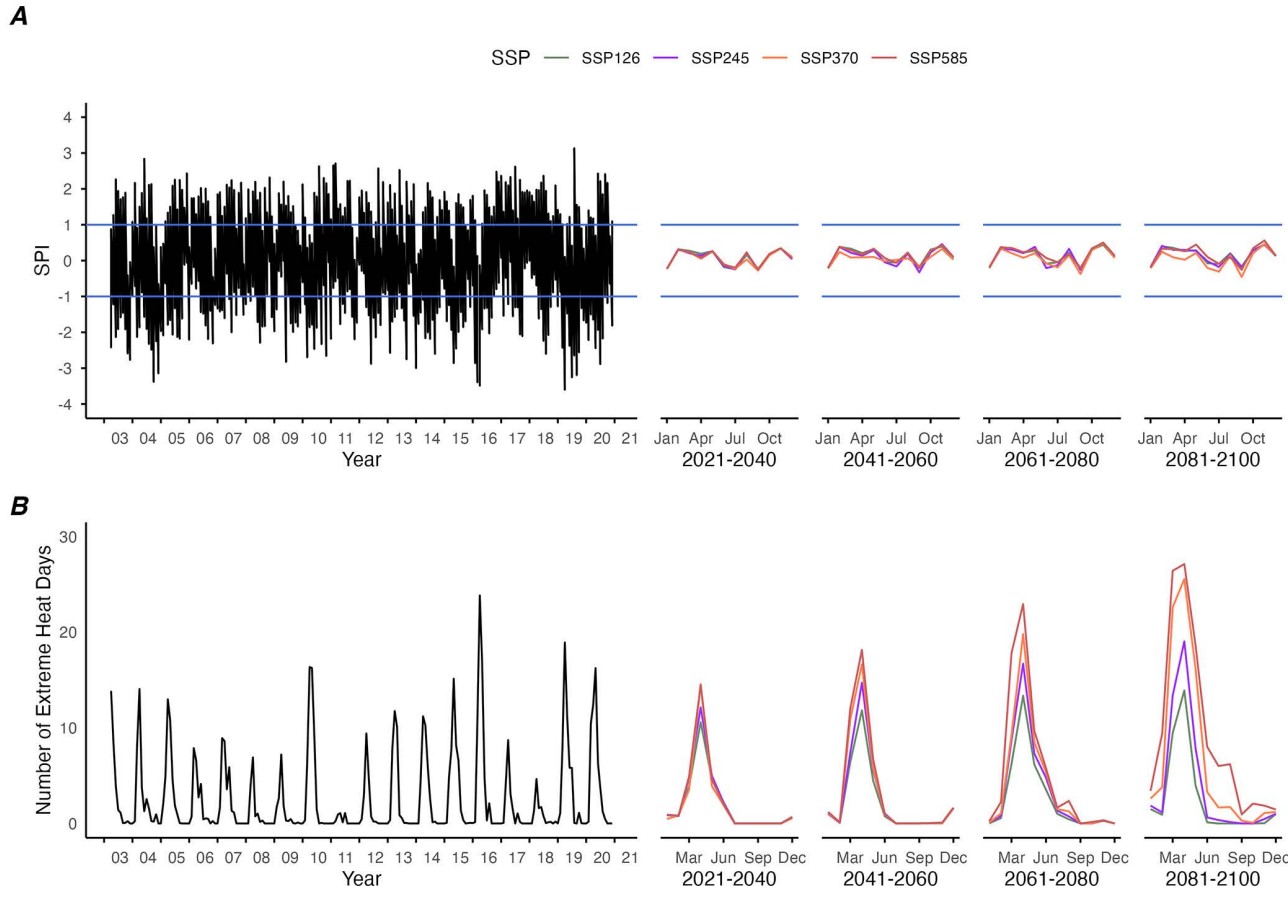

**Fig 1. Historical and projected monthly SPI and extreme heat days. (A)** Monthly SPI and **(B)** monthly extreme heat days from 2003 to 2021 and during the periods 2021–2040, 2041–2060, 2061–2080, 2081–2100 with climate change scenarios SSP126, SSP245, SSP370 and SSP585 in Thailand. Climate projections are based on the MIROC6 GCM.

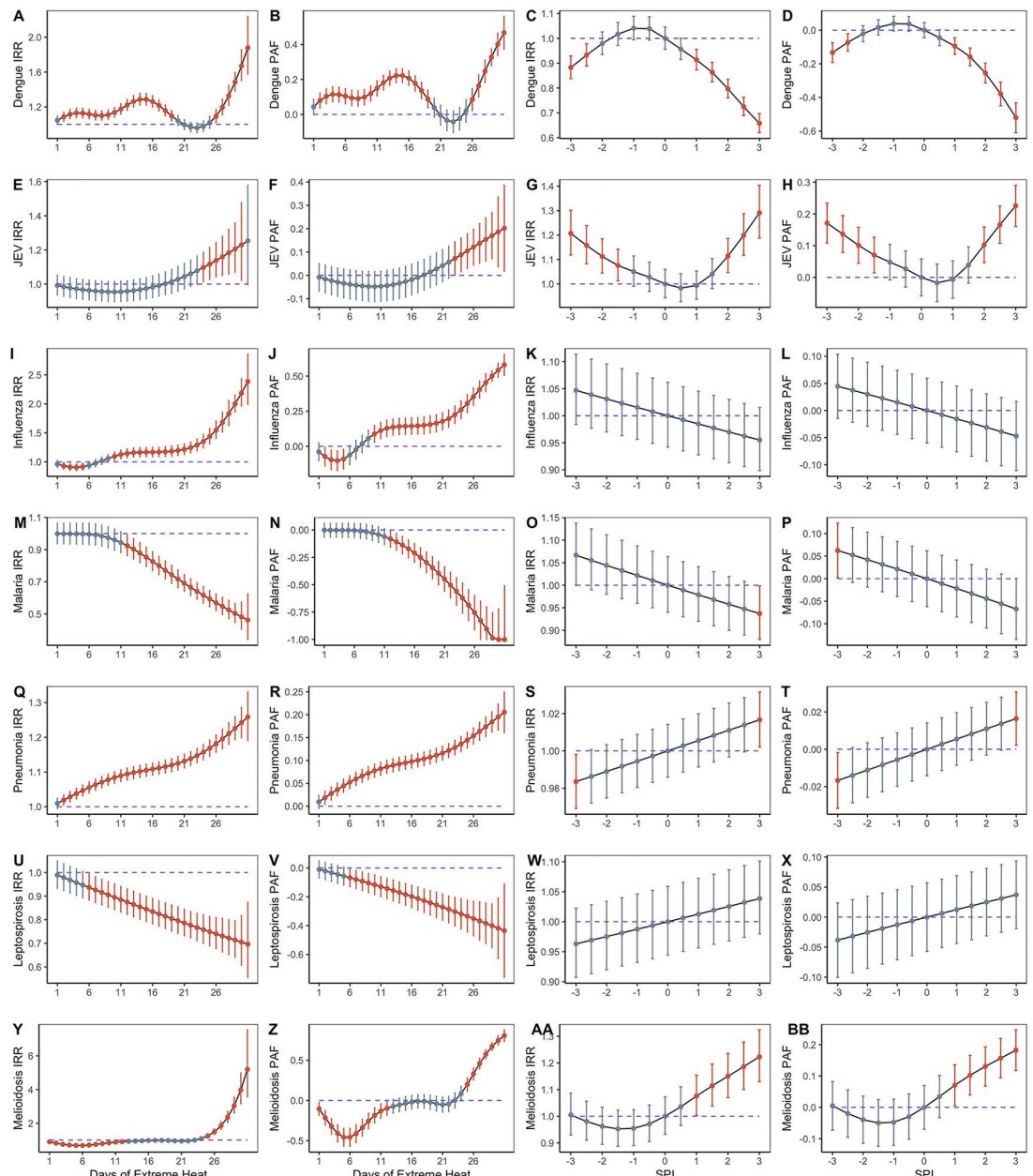

**Fig 2. National level incidence rate ratio (IRR) and population attributable fraction (PAF) over days of extreme heat and standardised precipitation index (SPI) for dengue, JEV, influenza, malaria, pneumonia, leptospirosis and melioidosis.** Bars represent 95% confidence intervals and red bars represent statistically significant points.

same trend, with positive values signifying that a proportion of cases in the population can be attributed to extreme heat, extreme dry weather and extreme wet weather respectively (S11–S17 Figs).

Extreme wet weather (i.e., SPI ≥1.5) was historically associated with lower incidence of dengue and higher incidence of JEV, pneumonia and melioidosis. At SPI levels of 3, dengue IRR was 0.66 (95% CI: 0.62, 0.69), while JEV IRR was 1.29 (95% CI: 1.19, 1.40). Extreme dry weather (i.e., SPI ≤ -1.5) was historically associated with higher incidence of JEV

and lower incidence of dengue and pneumonia (Fig 2). At SPI levels of 3, dengue IRR was 0.88 (95% CI: 0.83, 0.92), while JEV IRR was 1.21 (95% CI: 1.12, 1.30). Province-level IRRs and PAFs were concordant with these trends (S2-S24 Figs). Lag-exposure curves for SPI and extreme heat days can be found in S25 Fig.

## Overall change in disease risk during periods of extreme weather

Using the calibrated model, we projected the excess risk of infectious diseases in future climate change scenarios based on projections of extreme heat and rainfall from 2021 to 2100. Excess risk describes the changes in incidence rates with reference to the historical baseline from 2003 to 2020.

Pneumonia and melioidosis risks are expected to increase in the future during periods of extreme weather, particularly in the highest carbon emission scenario, SSP585 respectively (Figs 3D, 2L). Influenza risks during periods of extreme weather are highest in the moderate scenario SSP245, specifically in the first half of the century (Fig 3B). In other scenarios, there is conversely a decrease in influenza risk during periods of extreme weather (Figs 3A, 2C, 2D). A decrease in JEV risk from 2061–2080 is observed across all four climate change scenarios, except for SSP585, where there is an expected increased risk—though the excess risk still follows the same trend as other scenarios (Fig 3E-3H).

An increase in malaria risk is expected during periods of extreme weather across all climate change scenarios and time periods (Fig 3I-3L). The exception is in SSP245, which sees a decrease in 2021–2040 and 2081–2100 (Fig 3J). Conversely, a decrease in dengue risk is expected during periods of extreme weather, across all climate change scenarios and time periods, apart from SSP245 (Fig 3E-3H). Under SSP245, an increase in dengue risk is expected from 2021–2040, which is followed by an expected steep decrease in the rest of the century (Fig 3F). Similar to dengue, leptospirosis risk is expected to decrease during periods of extreme weather, across all climate change scenarios and time periods.

## Change in disease risk by individual extreme weather events

We stratified the projected change in disease risk by the type of extreme event – extreme heat, extreme dry weather and extreme wet weather. A decrease in dengue is expected during periods of extreme heat under all climate change scenarios other than SSP245 (Fig 4A). In SSP245, a large increase of 26.2% (95% CI: 0.90%, 51.7%) is expected from 2021–2040, which then declines and increases slightly again in 2081–2100 (Fig 4A). The decrease in dengue risk during extreme heat generally contradicts the historical association between extreme heat and increased dengue risk. We found that changes in dengue risk were primarily driven by future relative humidity. When relative humidity was held constant between the baseline and projection scenarios, dengue excess risk was small and statistically insignificant across all SSPs and time points (S26 Fig). This suggests that the projected decrease in dengue risk was likely driven by lower levels of relative humidity, which have been historically associated with reduced dengue risk (S27 Fig). Among other diseases, relative humidity was not found to drive any changes in excess risk during periods of extreme heat. A large decrease in dengue is expected in periods of extreme dry and wet weather across all climate change scenarios (Figs 4B, 3C). Extreme dry and wet weather was historically associated with decreases in dengue risk (Fig 2). The expected decrease in dengue risk during extreme dry weather ranges from -52.1% (95% CI: -63.2%, -41.0%) in 2081–2100 under SSP370 to -25.4% (95% CI: -41.0%, -9.77%) in 2021–2040 under SSP245 (Fig 4B). Dengue risk during extreme wet weather is expected to decrease the most by 52.9% (95% CI: 43.0%, 62.8%) in 2041–2060 under SSP370 and decreases the least by 35.0% (95% CI: 21.1%, 48.7%) in 2021–2040 under SSP245 (Fig 4C). For other diseases, this effect was more modest and statistically insignificant (Fig 4).

We expect JEV risk to decrease during periods of extreme dry and extreme wet weather (Fig 4E, 4F). The greatest decline is expected during periods of extreme dry weather in 2081–2100 under SSP245 (ER: -18.2%, 95% CI: -43.4%, 6.96%) (Fig 4E). Conversely, malaria risk is expected to increase during periods of extreme dry weather and wet weather, but more so during extreme dry weather (Fig 4H, 4I). This is given that there was a historical association between extreme dry weather and increased malaria risk (Fig 2).

PLOS **Neglected Tropical Diseases**

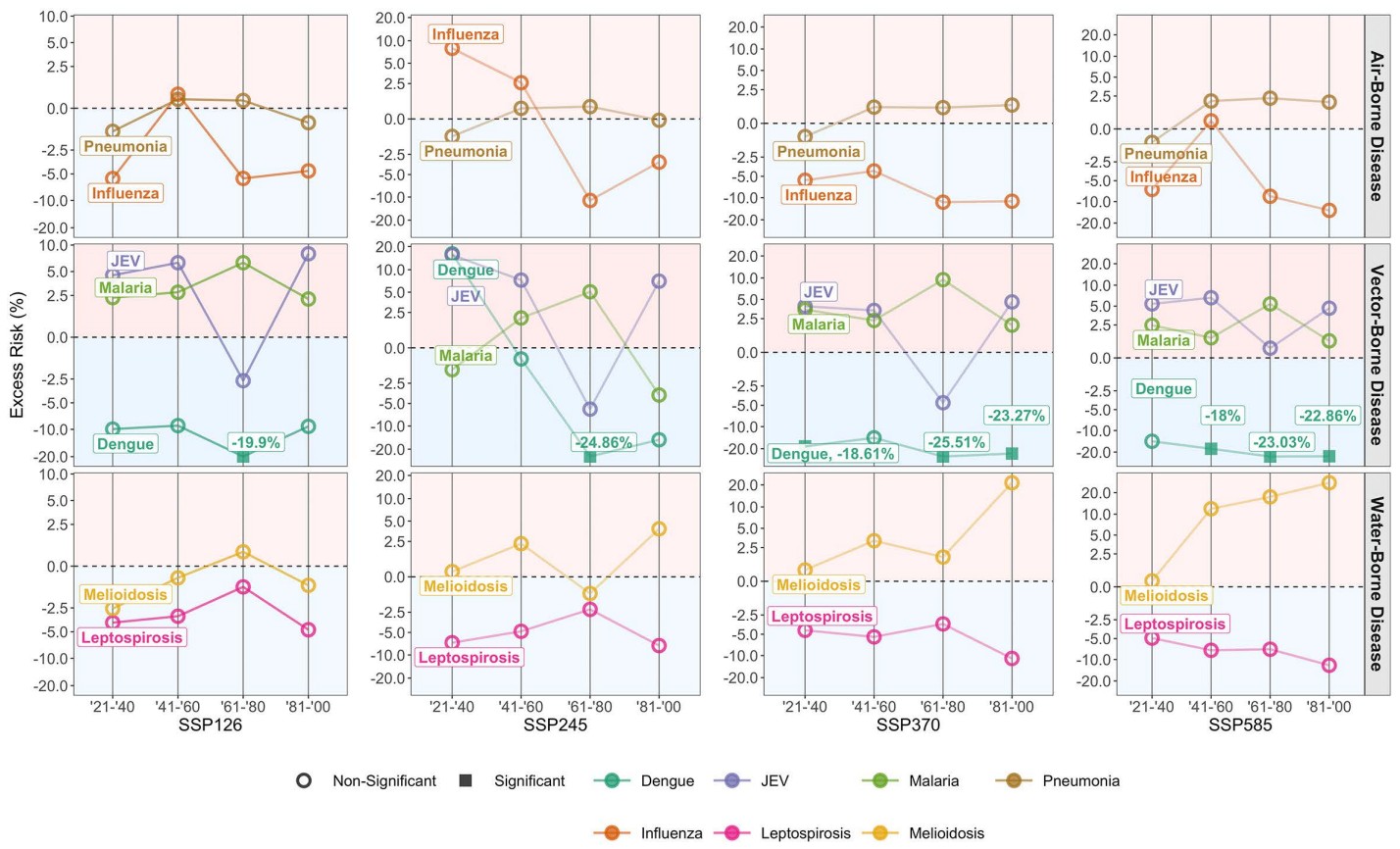

**Fig 3. Excess risk of pneumonia, influenza, JEV, malaria, dengue, melioidosis and leptospirosis during periods of extreme weather, across 2021 to 2100, in climate change scenarios SSP126, SSP245, SSP370 and SSP585.** Excess risk represents the percentage change in annual disease case counts from the historical baseline from 2003 to 2020.

Influenza risk is expected to increase during periods of extreme heat only under SSP245, which peaks in 2021–2040 (ER: 9.74%, 95% CI: -11.8%, 31.3%) (Fig 5A). Historically, having over 8 days of extreme heat in a month is associated with increased influenza risk (Fig 2). During periods of extreme dry weather, influenza risk is expected to decrease, particularly in SSP245 and SSP370 (Fig 5B). Under SSP245, influenza risk is expected to decrease during periods of extreme dry weather to the greatest extent from 2061–2080 (ER: -18.9%, 95% CI: -35.4%, -2.33%) (Fig 5B). The risk of influenza is expected to decrease more during periods of extreme wet weather than during periods of extreme dry weather (Fig 5B, 5C). The greatest expected decrease during periods of extreme wet weather can be seen under SSP370, which decreases by -37.8% (95% CI: -50.8%, -24.8%) from 2061–2080 (Fig 5C). This relationship is also seen historically, where extreme wet weather has a negative association with influenza risk (Fig 2).

The risk of melioidosis is projected to increase on extreme heat days, especially under higher carbon emission scenarios like SSP370 and SSP585 (Fig 5J). SSP585 shows the most pronounced increase over time, although the confidence intervals remain wide (Fig 5J). By 2100, melioidosis risk increases by 27.8% (95% CI: -39.4%, 94.9%) during periods of extreme heat (Fig 5J). A large increase in melioidosis is expected during periods of extreme wet weather, although confidence intervals are wide (Fig 5L). Broad confidence intervals could be explained by low monthly case counts across provinces. The greatest increase is expected under SSP585, from 2041–2060 (ER: 25.4%, 95% CI: -32.1%, 82.8%) (Fig 5L).

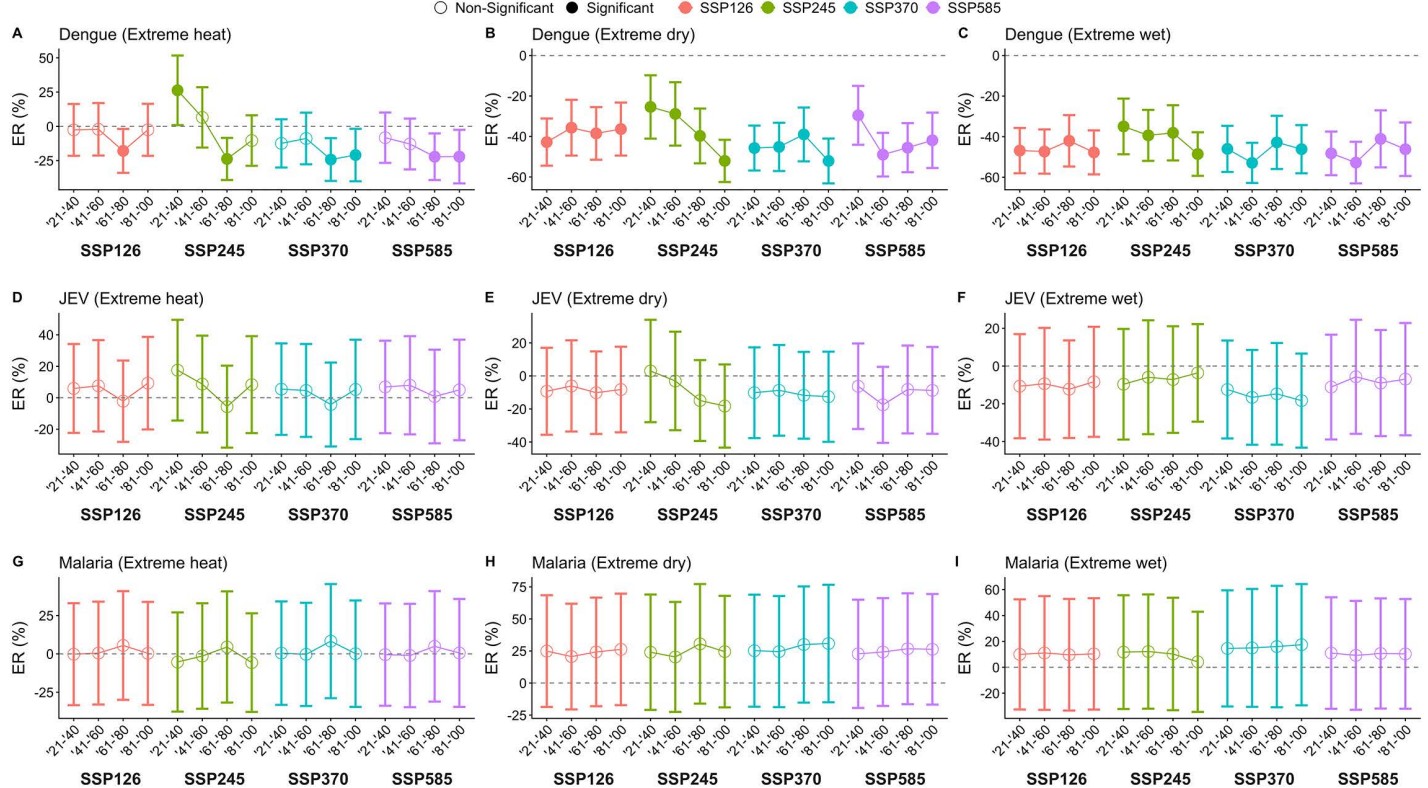

**Fig 4. Excess risk (ER) of JEV, malaria and dengue during periods of extreme heat, extreme dry weather and extreme wet weather, across 2021 to 2100, in climate change scenarios SSP126, SSP245, SSP370 and SSP585.** Excess risk represents the percentage change in annual disease case counts from the historical baseline from 2003 to 2020.

### Trends of dengue excess risk during extreme weather in SSP245

We used the calibrated model to estimate province-level changes in incidence rate during periods of extreme weather compared to the baseline of 2003–2019, which included periods of extreme heat, extreme dry weather and extreme wet weather. Province-level excess risks of all diseases during periods of extreme weather across climate change scenarios can be found in S28-S34 Figs, where it can be seen that influenza and dengue have the most statistically significant province-level excess risks. We conducted a finer analysis on the change in dengue risk during periods of extreme weather in SSP245, which showed a significant increase in the first half of the century (Fig 3F). Moreover, SSP245 is generally considered the most likely scenario [49]. We focused on the excess risks of five provinces which had the highest dengue burden from 2003 to 2019 (Fig 6A).

From 2021–2040, dengue risk is expected to increase in most provinces during periods of extreme weather (Fig 6B). Provinces with historically high dengue incidence, such as Bangkok, are expected to have an increased risk of dengue by 42.2% (95% CI: 13.6%, 70.7%) during periods of extreme weather (Fig 6A). On the other hand, there were also provinces which are expected to experience a decrease in dengue risk during extreme weather in this time, such as Chiang Mai (ER: -24.6%, 95% CI: -40.1%, -9.00%) (Fig 6A). From 2041–2060, a general decrease in excess risk is seen, particularly in the Southern region (Fig 6A, 6B). In contrast, provinces in central Thailand are expected to have increase in dengue risk (Fig 6B).

From 2061–2080, dengue risk during extreme weather are expected to decrease and reach its lowest point in most provinces (Fig 6A, 6B). A notable decrease in risk during periods of extreme weather is expected in Chiang Rai (ER: -34.2%,

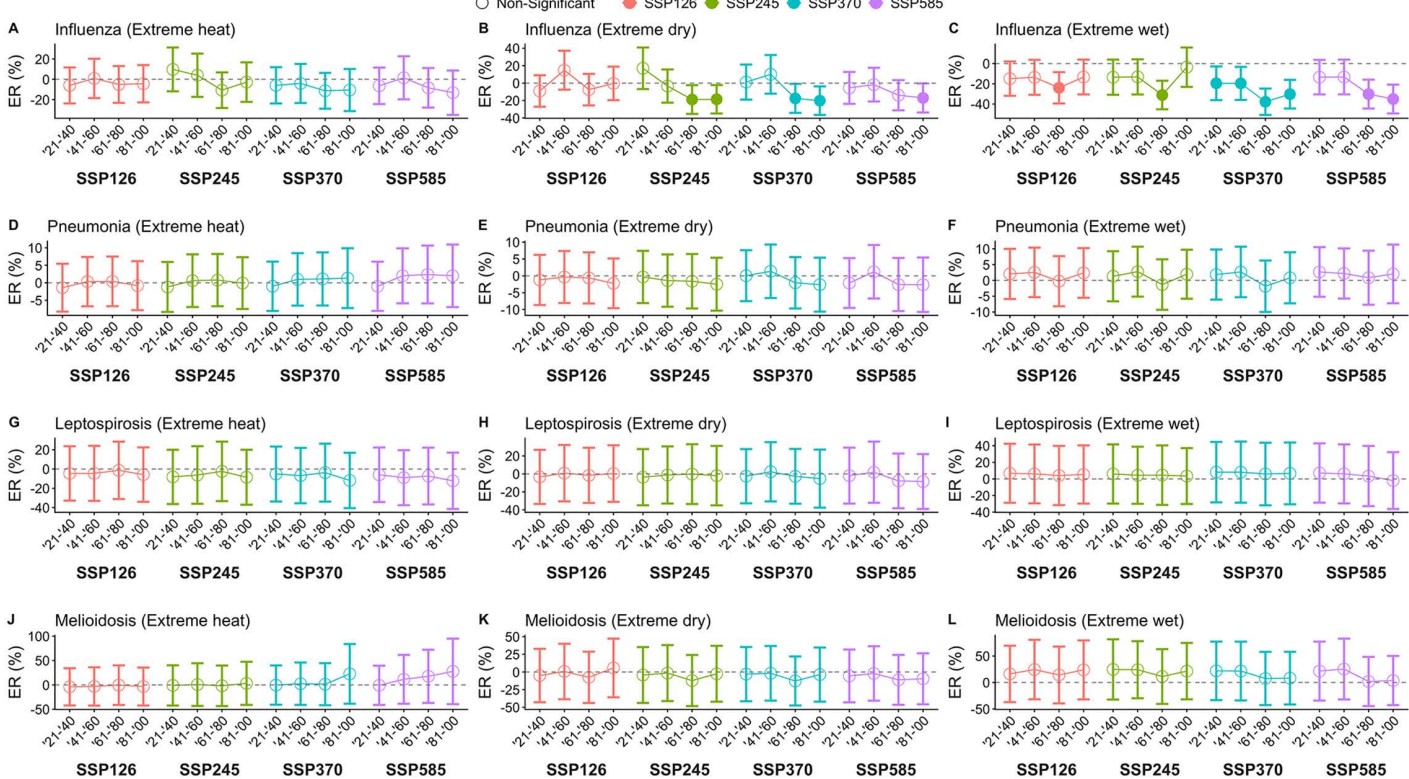

**Fig 5. Excess risk (ER) of influenza, pneumonia, leptospirosis and melioidosis during periods of extreme heat, extreme dry weather and extreme wet weather, across 2021 to 2100, in climate change scenarios SSP126, SSP245, SSP370 and SSP585.** Excess risk represents the percentage change in annual disease case counts from the historical baseline from 2003 to 2020.

95% CI: -47.2%, -21.1%) (Fig 6A). From 2081–2100, dengue risk remains moderately low among most provinces during periods of extreme weather (Fig 6A, 6B).

The increased dengue in central Thailand from 2021–2060 is likely to be driven by moderate levels of extreme heat in that period (Figs 4A-4C, S1, S35). Moreover, in the southern provinces with lower levels of extreme heat, no increase is expected in this period (S1 Fig). In the second half of the century, higher levels of extreme dry and extreme wet weather is expected to drive decreases in dengue across most of the provinces (Figs 4A-4C, S1). This aligns with our observation that extreme wet or dry weather was historically associated with a lower incidence rate of dengue (Fig 2).

### Trends of influenza excess risk during extreme weather in SSP245

Similarly, we narrowed our analysis to the change in influenza risk during periods of extreme weather in SSP245, which is expected to experience significant changes at the province level over the century (Fig 7A, 7B).

From 2021–2040, an increase in influenza is expected in most provinces (Fig 7B). The heterogenous increase across provinces is likely to be driven by moderate levels of extreme heat and extreme dry weather in the area (Figs 5A-5C, S1). Influenza risk is expected to peak in 2041– 2060, where most provinces is expected to have an increased risk during periods of extreme weather, particularly in Northeastern and Central Thailand. Nakhon Ratchasima, a province with historically high influenza burden, is expected to have an excess risk of 36.8% (95% CI: 9.83%, 63.8%) during periods of extreme weather (Fig 7A). This is likely driven by increasing levels of extreme heat in those regions, which was historically associated with increased influenza risk (Figs 2, S1, S36). The lowest levels of influenza excess risk during extreme

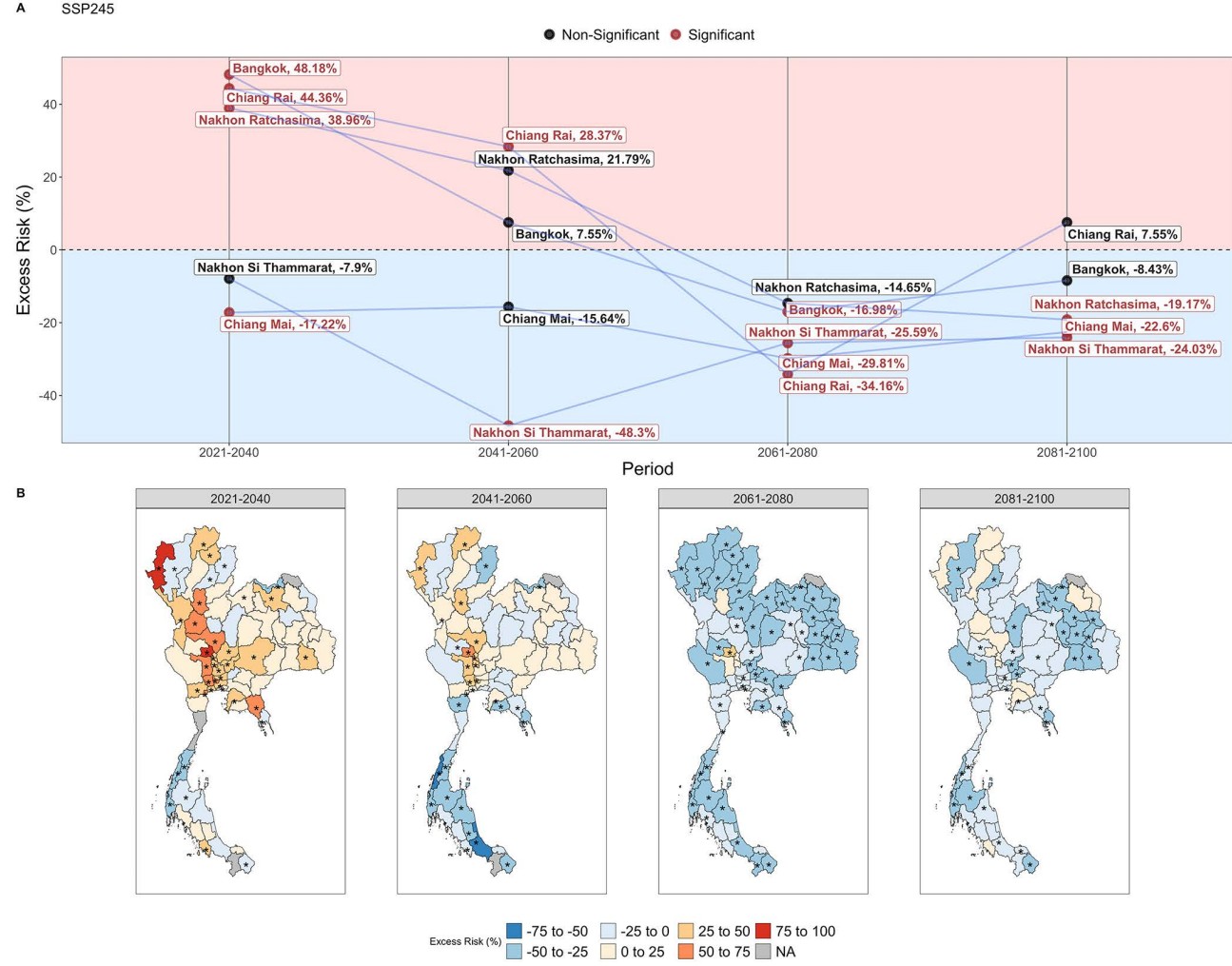

**Fig 6.** (A) Excess risk of the five provinces with the historically highest incidence of dengue during periods of extreme weather, under SSP245 from 2021–2100. (B) Province-level excess risks of dengue under SSP245 from 2021–2100. Asterisks indicate a statistically significant excess risk value. Excess risk represents the percentage change in annual disease case counts from the historical baseline from 2003 to 2020. Map created using GADM data (https://gadm.org/index.html, freely available for academic use). The map outlines and administrative boundaries are used with permission for academic publishing.

weather across the five highlighted provinces are seen from 2041–2060, particularly in Southern Thailand (Fig 7B). From 2081– 2100, influenza risks across the five highlighted provinces will reach a generally moderate level where about half of provinces are expected to experience an increase in influenza risk during extreme weather periods while the other half are expected to experience a decrease (Fig 7B). The lower levels of influenza, particularly in Southern Thailand, are likely to be driven by extreme wet weather which was historically associated with lower influenza risk (Figs 2, 5A–5C, S1).

## Model evaluation

We compared the AIC of the GAM models to their corresponding GLM models, which used the same covariates without the smooth function. The AIC of each GAM model was lower than its corresponding GLM model in all 7 models (S3 Table). This showed that the incorporation of smooth functions to model predictors improved model fit in almost all cases.

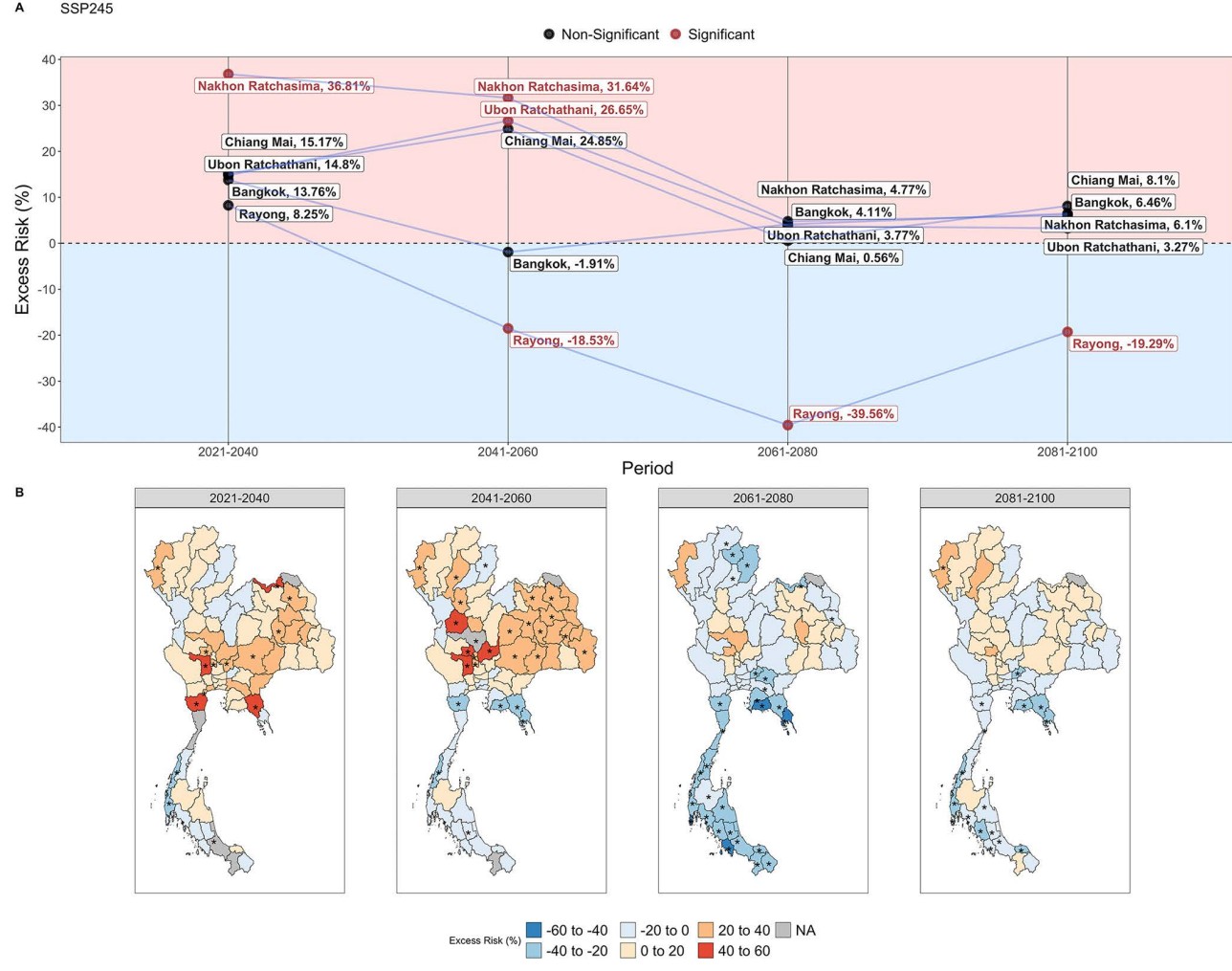

**Fig 7.** (A) Excess risk of the five provinces with the historically highest incidence of influenza during periods of extreme weather, under SSP245 from 2021–2100. (B) Province-level excess risks of influenza under SSP245 from 2021–2100. Asterisks indicate a statistically significant excess risk value. Excess risk represents the percentage change in annual disease case counts from the historical baseline from 2003 to 2020. Map created using GADM data (https://gadm.org/index.html, freely available for academic use). The map outlines and administrative boundaries are used with permission for academic publishing.

We evaluated an alternative GAM model which included future population as an offset and future population density as a predictor instead of current population. Predictions were heavily skewed by future population projections (S37–S43 Figs). This was most likely due to models being unable to handle population projections which exceed current observations. Moreover, we wanted to observe the change in risk solely from extreme weather and not population levels. Hence, in this study, we assumed that population would remain at current levels based on data from 2003 to 2021.

## Sensitivity analysis

We estimated the excess risks attributed to extreme weather events based on 2 other global climate models (GCMs), CMCC-ESM2 and IPSL-CM6A-LR to capture the uncertainty inherent in the climate projections and test the robustness of our findings. We also estimated excess risks based on a multi-model GCM ensemble, calculated as the mean of MIROC6,

CMCC-ESM2 and IPSL-CM6A-LR outputs. Across all the diseases studied, the general trends of excess risk attributable to each extreme weather scenario based on the alternative GCMs were similar to the main projections in this study (S44-S50 Figs). We evaluated the fit of GAM models without including relative humidity as a predictor across provinces and diseases to determine if inclusion of relative humidity would enhance model fit. Including relative humidity improved the fit of all 7 models, given the AIC of all models with relative humidity was lower than the AIC of models without relative humidity (S4 Table). Moreover, we evaluated the fit of GAM models using a cubic regression spline instead of thin plate regression spline across provinces and diseases and found that the difference in spline used did not change model fit in most models (S5 Table).

We also estimated the excess risk attributable to extreme weather sub-scenarios while controlling for the presence or absence of other concurrent extreme weather conditions. Certain extreme weather events, such as extreme heat and extreme dry weather, may tend to take place simultaneously [48]. Thus, we explored if the excess risks atrributable to one type of extreme weather might actually reflect combined effects.

We first compared excess risk attributable to extreme heat against sub-scenarios combining extreme heat and extreme dry weather, extreme wet weather or normal conditions. Across diseases, the excess risks attributable to extreme heat were similar to excess risks attributable to extreme heat with no extreme dry or wet weather (S51 Fig). This suggests that excess risks attributable to extreme heat estimated were not influenced by other extreme weather conditions.

Additionally, we compared the excess risk attributable to extreme dry weather with those under sub-scenarios involving extreme dry weather with or without extreme heat respectively. Excess risks attributable to extreme dry weather resembled the excess risks attributable to extreme dry weather with no extreme heat (S52 Fig). This indicates that the excess risks attributable to extreme dry weather were not confounded by the effects of extreme heat.

Lastly, we compared the excess risk attributable to extreme wet weather with those under sub-scenarios involving extreme wet weather with or without extreme heat respectively. The excess risks were likewise consistent, suggesting that the excess risks attributable to extreme wet weather were similarly not confounded by the effects of extreme heat (S53 Fig).

## Discussion

Our study projected the change in risks of vector-borne diseases, air-borne diseases and water-borne diseases from historical baselines during periods of extreme weather in Thailand. We project a decrease in future dengue risk during extreme weather events in all climate change scenarios. The exception was in scenario SSP245, which saw a large increased risk from 2021-2060. This increased risk was observed in northern and central regions of Thailand which is projected to face higher levels of extreme heat than other regions. This aligns with our finding that moderate levels of extreme heat were historically associated with higher dengue incidence. The increased risk is expected to be followed by a decrease in the second half of the century, likely due to extreme dry and wet conditions and lower relative humidity which were historically associated with lower dengue incidence. We also project an increase in influenza risk during extreme weather events under SSP245, particularly in the first half of the century. The increase is driven by extreme heat and dry weather in Northeastern and Central Thailand, which were found to be historically associated with increased influenza risk in these provinces. We observed a decrease in influenza risk in the second half of the century, which could be attributed to extreme wet weather conditions. These extreme weather events were found to be associated with lower influenza incidence in those provinces. The projected increases in disease risk are a cause for concern especially given that SSP245 is considered the baseline scenario that is most likely to take place [49].

The lower levels of influenza, particularly in Southern Thailand, is likely attributed to extreme wet and dry weather (Figs 5A–5C, S1). Moreover, the waning influenza risk in Northeastern and Central Thailand is likely due to high levels of extreme heat (i.e., above 27 days of extreme heat in a month) which is associated with lower levels of influenza risk.

 

Under the SSP245 scenario, dengue is expected to rise from 2021–2040. High-risk provinces in northern and central Thailand, particularly Bangkok, Chiang Rai, Ratchaburi, Nakhon Ratchasima, and Si Sa Ket, should prioritize vector control and hospital preparedness during heatwaves accompanied by normal rainfall, as both dry and wet extremes appear to be protective. Influenza is also expected to rise from 2021–2060 under SSP245. High-risk provinces such as Nakhon Ratchasima, Buri Ram, Ubon Ratchathani, and Chiang Mai should strengthen surveillance, vaccination campaigns, and healthcare surge capacity during heatwaves coupled with extended dry periods, which are both associated with higher transmission risks. These operational triggers may inform region-specific preparedness and response planning under future climate scenarios.

The clearest relation between extreme weather and infectious disease risk is observed in vector-borne diseases, given the direct influence of weather on the behaviour and reproduction of vectors [50]. Among the 3 vector-borne diseases, dengue was affected by extreme weather in most future scenarios. This could be because in Thailand, dengue transmission is driven largely by climate factors while malaria transmission is driven largely by levels of forest disturbance and JEV outbreaks are strongly associated with amplifying hosts like pigs [44,51,52]–these were factors which were not considered in our model.

Under SSP245, dengue is expected to first increase with moderate levels of extreme heat. This can be explained by the shortening of the extrinsic incubation period with increasing temperatures [41]. Transmission rate increases from the reduction in the time taken for *Aedes* mosquitoes to become infectious, leading to higher dengue incidence. In the second half of the century, dengue is expected to decrease in periods of extreme dry weather and extreme wet weather. Associations between dengue risk, extreme wet and extreme dry conditions have been observed in other studies [53]. *Aedes aegypti*, the main vector of dengue, breeds in a wide variety of natural and artificial water containers [54]. In months of drought, the number of potential breeding sites for laying eggs reduces, hence restricting the *Aedes aegypti* population and decreasing transmission of dengue. On the other hand, excessive rainfall could wash out breeding sites and hence reduce transmission of dengue [55]. Decrease in dengue risk can also be driven by a decrease in relative humidity. Dengue transmission is affected by lower relative humidity, which can shorten the lifespan of mosquitoes and reduce oviposition rates [56,57].

Influenza risk is first expected to increase from historical baselines in periods of extreme heat and extreme dry weather under SSP245. This could be because physiological stress from heat and dehydration weakens respiratory immunity [45,58]. Furthermore, in Thailand, influenza cases were historically found to decline following hot and dry months [59]. These findings support our results of decreased influenza risk during extreme dry weather. Influenza is then expected to decrease later in the century during periods of extreme wet weather, despite "humid-rainy" seasons being commonly positively associated with influenza risk [59–61]. This may be a result of decreased human mobility and social mixing from extreme wet weather, reducing influenza transmission.

In our spatiotemporal study, we found that some changes in disease risk compared to baseline levels appeared minimal at the national resolution but had more pronounced patterns at the provincial resolution. For example, the change in influenza risk showed mild changes when risk was aggregated across provinces. However, at the province level, we observed heterogenous risk across provinces, with some provinces experiencing large increases and some experiencing decreases in influenza risk. This phenomenon can be attributed to the heterogenous projected temperature and precipitation across provinces. Northern Thailand is projected to experience more extreme dry weather than the rest of Thailand, while Central and Eastern Thailand is projected to experience more extreme heat. Hence, increases in influenza in SSP245 during periods of extreme weather are projected mostly in Northern, Central and Eastern Thailand, while increases in dengue in SSP245 during periods of extreme weather are projected mostly in Central and Eastern Thailand. This underscores the importance of analysing these regional differences. The heterogeneity in risk across provinces could be tied to factors like differences in the degree of extreme weather projected across provinces, or differences in how meteorological factors are associated with disease risk in each province.

To the authors' best knowledge, this is the first study to project the future risks of multiple infectious diseases associated with future extreme weather events. Our study had several strengths. Firstly, we studied a wide range of seven diseases that vary in transmission mode, which allowed us to have a broad understanding of how extreme weather affects infectious diseases. Secondly, the use of tailored indicators for extreme weather events rather than raw meteorological variables allowed for a more precise assessment of the excess risks attributable to extreme weather. Additionally, we used a long time series dataset spanning nearly two decades, which provided a solid basis for examining epidemiological trends over time. This extended timeframe allowed us to capture both seasonal variations and longer-term shifts, offering a clearer picture of changes in climate patterns. Furthermore, we validated our approach by calibrating our model across different model structures, such as using varying lags and covariates. Lastly, we conducted our analysis at the province level, which revealed the spatial heterogeneity in the projected excess disease risks within Thailand due to different projected extreme heat or precipitation conditions.

A limitation of our study was in the temporal disaggregation of the climate change projections. Future monthly temperature was disaggregated to daily temperature to determine the number of extreme heat days. However, this approach may not accurately capture the true variability and daily extremes within each month. Hence, the projected frequency of extreme heat events may differ slightly from actual conditions. Moreover, we did not apply additional bias correction to the GCM inputs, so weather extremes may have been under-estimated or over-estimated, which could influence the estimated impact of extreme weather on disease outcomes. Meanwhile, extreme SPI values appeared less frequently in the GCM projections compared to the historical period. This is likely because each monthly value represents a 20-year average, which smooths out year-to-year extremes. Hence, our results might underestimate the impact of extreme dry or wet weather on disease risk. Future research should focus on the development of GCM outputs that provide continuous monthly data for future periods, which would allow for a more accurate representation of interannual climate variability and extreme events. Additionally, as the focus of our study was extreme weather and due to the lack of future projected data under different SSPs, we did not account for variables such as changes in land use, use of intervention and proximity to amplifying hosts that may drive the transmission of diseases like JEV and malaria.

We were unable to include a autocorrelation structure in our model due to convergence issues arising from the combination of lagged covariates and overdispersed count outcomes. We also could not account for future population growth as future population exceeded historical maxima, hence incorporating future population in projections led to implausibly large and uncertain estimates. Furthermore, we did not account for potential changes in reporting rates over time. Without adjusting for these factors, there is a risk that fluctuations in reported cases could be influenced by reporting practices rather than true epidemiological patterns. Short-term events, such as influenza pandemics or other disruptions to surveillance, may have temporarily affect reported case counts. However, their impact on long-term trends is likely minimal given the length of the study period. We also were unable to incorporate population immunity dynamics and future interventions in our model. This would require refining of our predictions in the future when such data is available.

Overall, we found that generally, dengue and influenza risk is expected to decrease from historical baselines during periods of extreme weather. We observed that each disease was projected at different trends and magnitudes with varying climate change-related extreme weather events. At the province level, expected changes in disease risk during periods of extreme weather are heterogeneous. This warrants targeted public health interventions at a finer administrative level to adapt to increases in disease risk as a result of extreme weather events. Although extreme heat, extreme dry and extreme weather were associated with decreased risk of some infectious diseases, we should remain vigilant against their impacts on human health, given its probable increase in intensity as a result of climate change [62]. Further work is needed to assess the projected impact of extreme weather on infectious diseases in other climatic and socioeconomic contexts. Our methodological framework developed here could be used to evaluate risks in temperate climates and in communities with varying health infrastructure, adaptive capacities, and economic resilience. Such work would provide a broader understanding of how climate-driven health risks manifest globally and inform region-specific adaptation strategies.

## Supporting information

**S1 Table. Overdispersion test.** Cameron and Trivedi test was done on the case counts of each disease respectively to check if the data was overdispersed. Overdispersed data warrants the use of the negative binomial distribution to count data.
(DOCX)

**S2 Table. Model selection criteria.** Disease-specific generalized additive models were trained taking lagged extreme heat days, lagged standardized precipitation index (SPI), relative humidity and population density as variables. Thin plate splines were used to model non-linear relationships between lagged extreme heat days, lagged SPI and monthly disease case counts. Models including different number of lags were assessed and compared based on the Akaike information criterion (AIC), with lower values indicating better model fit to observations while penalizing for model complexity. The number of variable lags selected for models of each disease are bolded and underlined.
(DOCX)

**S3 Table. AIC of the GAM and GLM models for each disease.** Disease-specific generalized additive models (GAM) and generalised linear models (GLM) were trained with relative humidity, lagged extreme heat days, lagged standardized precipitation index (SPI) and population density as variables. The Akaike information criterion (AIC) of each model was calculated to compare model fit of GLM against GAM.
(DOCX)

**S4 Table. AIC of each model with and without relative humidity as a predictor.** Disease-specific generalized additive models (GAM) were trained with and without relative humidity (RH) respectively, together with lagged extreme heat days, lagged standardized precipitation index (SPI) and population density as variables. The lags chosen for extreme heat days and SPI are indicated in Supplementary Table S2. Thin plate splines were used to model non-linear relationships between lagged extreme heat days, lagged SPI and monthly disease case counts. The Akaike information criterion (AIC) of each model was calculated to compare model fit with inclusion of relative humidity as a predictor. AIC is used to assess model fit as it balances goodness of fit with model complexity. Lower AIC values which indicated better model fit are bolded.
(DOCX)

**S5 Table. AIC of GAM models using thin plate regression spline and cubic regression spline.** Disease-specific generalized additive models (GAM) were trained with lagged extreme heat days, lagged standardized precipitation index (SPI), relative humidity and population density as variables. The lags chosen for extreme heat days and SPI are indicated in Supplementary Table S2. Splines were used to model non-linear relationships between lagged extreme heat days, lagged SPI and monthly disease case counts. We explored whether using different splines would affect model fit. Province and disease-specific generalized additive models (GAM) were trained with thin plate regression splines (TR) and compared with GAM models using the same predictors and cubic regression splines (CR). The Akaike information criterion (AIC) of each model was calculated to compare model fit using different splines. AIC is used to assess model fit as it balances goodness of fit with model complexity. Lower AIC values which indicated better model fit are bolded.
(DOCX)

**S6 Table. Annual national projected excess risk of each disease attributable to overall extreme weather, extreme heat, extreme dry and extreme wet weather across time periods and climate change scenarios.** Disease-specific generalized additive models (GAM) were trained on historical data from 2003-2019 and used to project future case counts of the respective disease across 4 time periods (2021–2040, 2041–2060, 2061–2080, 2081–2100) and 4 climate change scenarios (SSP126, SSP245, SSP370, SSP585) during periods of extreme weather. National-level excess risk was calculated using the mean disease case counts across the historical period and the projected case counts at a respective

time period and climate change scenario. Excess risk represents the percentage change in disease cases compared to historical levels. Values in asterisk are statistically significant.
(DOCX)

**S1 Fig. Days of extreme heat and SPI across climate change scenarios by province under the MIROC6 general circulation model.** Monthly maximum, minimum temperature and total precipitation from the MIROC6 general circulation model was interpolated to daily frequency. Total precipitation was converted to Standardised Precipitation Index (SPI) and the number of extreme heat days were counted from the daily maximum temperature. An extreme heat day is one where the maximum temperature exceeds the 90th percentile of historical national maximum temperature, for a period of three or more days. Map created using GADM data (https://gadm.org/index.html, freely available for academic use). The map outlines and administrative boundaries are used with permission for academic publishing.
(PNG)

**S2 Fig. Days of extreme heat and SPI across climate change scenarios by province under the IPSL-CM6A-LR general circulation model.** Monthly maximum, minimum temperature and total precipitation from the IPSL-CM6A-LR general circulation model was interpolated to daily frequency. Total precipitation was converted to Standardised Precipitation Index (SPI) and the number of extreme heat days were counted from the daily maximum temperature. An extreme heat day is one where the maximum temperature exceeds the 90th percentile of historical national maximum temperature, for a period of three or more days. Map created using GADM data (https://gadm.org/index.html, freely available for academic use). The map outlines and administrative boundaries are used with permission for academic publishing.
(PNG)

**S3 Fig. Days of extreme heat and SPI across climate change scenarios by province under the CMCC-ESM2 general circulation model.** Monthly maximum, minimum temperature and total precipitation from the CMCC-ESM2 general circulation model was interpolated to daily frequency. Total precipitation was converted to Standardised Precipitation Index (SPI) and the number of extreme heat days were counted from the daily maximum temperature. An extreme heat day is one where the maximum temperature exceeds the 90th percentile of historical national maximum temperature, for a period of three or more days. Map created using GADM data (https://gadm.org/index.html, freely available for academic use). The map outlines and administrative boundaries are used with permission for academic publishing.
(PNG)

**S4 Fig. Incidence rate ratio of dengue over days of extreme heat (EH) for each province.** Fig show incidence rate ratio (IRR) of each disease over days of extreme heat for each province. The IRR gives the ratio of predicted cases with exposure to predicted cases with no exposure obtained from the disease-specific generalised additive models. The IRR for extreme heat gives the ratio between incidence rates at non-zero days of extreme heat in a month and the incidence rate with no days of extreme heat. An IRR above 1 represents increased incidence rate with exposure compared to no exposure, while an IRR below 1 represents decreased incidence rate. Points represent point estimates for the IRR with accompanying 95% confidence intervals. IRRs were derived using the ratio of predicted cases with exposure and predicted cases without exposure. Statistical significance was denoted with orange points when 95% CIs of the IRRs do not cross 1.
(JPEG)

**S5 Fig. Incidence rate ratio of Japanese encephalitis virus over days of extreme heat (EH) for each province.** Fig show incidence rate ratio (IRR) of each disease over days of extreme heat for each province. The IRR gives the ratio of predicted cases with exposure to predicted cases with no exposure obtained from the disease-specific generalised additive models. The IRR for extreme heat gives the ratio between incidence rates at non-zero days of extreme heat in

a month and the incidence rate with no days of extreme heat. An IRR above 1 represents increased incidence rate with exposure compared to no exposure, while an IRR below 1 represents decreased incidence rate. Points represent point estimates for the IRR with accompanying 95% confidence intervals. IRRs were derived using the ratio of predicted cases with exposure and predicted cases without exposure. Statistical significance was denoted with orange points when 95% CIs of the IRRs do not cross 1.

(JPEG)

**S6 Fig. Incidence rate ratio of influenza over days of extreme heat (EH) for each province.** Fig show incidence rate ratio (IRR) of each disease over days of extreme heat for each province. The IRR gives the ratio of predicted cases with exposure to predicted cases with no exposure obtained from the disease-specific generalised additive models. The IRR for extreme heat gives the ratio between incidence rates at non-zero days of extreme heat in a month and the incidence rate with no days of extreme heat. An IRR above 1 represents increased incidence rate with exposure compared to no exposure, while an IRR below 1 represents decreased incidence rate. Points represent point estimates for the IRR with accompanying 95% confidence intervals. IRRs were derived using the ratio of predicted cases with exposure and predicted cases without exposure. Statistical significance was denoted with orange points when 95% CIs of the IRRs do not cross 1.

(JPEG)

**S7 Fig. Incidence rate ratio of leptospirosis over days of extreme heat (EH) for each province.** Fig show incidence rate ratio (IRR) of each disease over days of extreme heat for each province. The IRR gives the ratio of predicted cases with exposure to predicted cases with no exposure obtained from the disease-specific generalised additive models. The IRR for extreme heat gives the ratio between incidence rates at non-zero days of extreme heat in a month and the incidence rate with no days of extreme heat. An IRR above 1 represents increased incidence rate with exposure compared to no exposure, while an IRR below 1 represents decreased incidence rate. Points represent point estimates for the IRR with accompanying 95% confidence intervals. IRRs were derived using the ratio of predicted cases with exposure and predicted cases without exposure. Statistical significance was denoted with orange points when 95% CIs of the IRRs do not cross 1.

(JPEG)

**S8 Fig. Incidence rate ratio of malaria over days of extreme heat (EH) for each province.** Fig show incidence rate ratio (IRR) of each disease over days of extreme heat for each province. The IRR gives the ratio of predicted cases with exposure to predicted cases with no exposure obtained from the disease-specific generalised additive models. The IRR for extreme heat gives the ratio between incidence rates at non-zero days of extreme heat in a month and the incidence rate with no days of extreme heat. An IRR above 1 represents increased incidence rate with exposure compared to no exposure, while an IRR below 1 represents decreased incidence rate. Points represent point estimates for the IRR with accompanying 95% confidence intervals. IRRs were derived using the ratio of predicted cases with exposure and predicted cases without exposure. Statistical significance was denoted with orange points when 95% CIs of the IRRs do not cross 1.

(JPEG)

**S9 Fig. Incidence rate ratio of melioidosis over days of extreme heat (EH) for each province.** Fig show incidence rate ratio (IRR) of each disease over days of extreme heat for each province. The IRR gives the ratio of predicted cases with exposure to predicted cases with no exposure obtained from the disease-specific generalised additive models. The IRR for extreme heat gives the ratio between incidence rates at non-zero days of extreme heat in a month and the incidence rate with no days of extreme heat. An IRR above 1 represents increased incidence rate with exposure compared to no exposure, while an IRR below 1 represents decreased incidence rate. Points represent point estimates for the IRR with

accompanying 95% confidence intervals. IRRs were derived using the ratio of predicted cases with exposure and predicted cases without exposure. Statistical significance was denoted with orange points when 95% CIs of the IRRs do not cross 1. (JPEG)

**S10 Fig. Incidence rate ratio of pneumonia over days of extreme heat (EH) for each province.** Fig show incidence rate ratio (IRR) of each disease over days of extreme heat for each province. The IRR gives the ratio of predicted cases with exposure to predicted cases with no exposure obtained from the disease-specific generalised additive models. The IRR for extreme heat gives the ratio between incidence rates at non-zero days of extreme heat in a month and the incidence rate with no days of extreme heat. An IRR above 1 represents increased incidence rate with exposure compared to no exposure, while an IRR below 1 represents decreased incidence rate. Points represent point estimates for the IRR with accompanying 95% confidence intervals. IRRs were derived using the ratio of predicted cases with exposure and predicted cases without exposure. Statistical significance was denoted with orange points when 95% CIs of the IRRs do not cross 1. (JPEG)

**S11 Fig. Incidence rate ratio of dengue over days of SPI for each province.** Fig shows incidence rate ratio (IRR) of each disease over days of SPI for each province. The IRR gives the ratio of predicted cases with exposure to predicted cases with no exposure obtained from the disease-specific generalised additive models. Specifically, the IRR for SPI gives the ratio between incidence rates at varying values of SPI and the incidence rate at 0 SPI. An SPI value of 0 represents average precipitation conditions based on the long-term mean precipitation in that location. An IRR above 1 represents increased incidence rate with exposure compared to no exposure, while an IRR below 1 represents decreased incidence rate. Points represent point estimates for the IRR with accompanying 95% confidence intervals. IRRs were derived using the ratio of predicted cases with exposure and predicted cases without exposure. Statistical significance was denoted with orange points when 95% CIs of the IRRs do not cross 1. (JPEG)

**S12 Fig. Incidence rate ratio of Japanese encephalitis virus over days of SPI for each province.** Fig shows incidence rate ratio (IRR) of each disease over days of SPI for each province. The IRR gives the ratio of predicted cases with exposure to predicted cases with no exposure obtained from the disease-specific generalised additive models. Specifically, the IRR for SPI gives the ratio between incidence rates at varying values of SPI and the incidence rate at 0 SPI. An SPI value of 0 represents average precipitation conditions based on the long-term mean precipitation in that location. An IRR above 1 represents increased incidence rate with exposure compared to no exposure, while an IRR below 1 represents decreased incidence rate. Points represent point estimates for the IRR with accompanying 95% confidence intervals. IRRs were derived using the ratio of predicted cases with exposure and predicted cases without exposure. Statistical significance was denoted with orange points when 95% CIs of the IRRs do not cross 1. (JPEG)

**S13 Fig. Incidence rate ratio of influenza over days of SPI for each province.** Fig shows incidence rate ratio (IRR) of each disease over days of SPI for each province. The IRR gives the ratio of predicted cases with exposure to predicted cases with no exposure obtained from the disease-specific generalised additive models. Specifically, the IRR for SPI gives the ratio between incidence rates at varying values of SPI and the incidence rate at 0 SPI. An SPI value of 0 represents average precipitation conditions based on the long-term mean precipitation in that location. An IRR above 1 represents increased incidence rate with exposure compared to no exposure, while an IRR below 1 represents decreased incidence rate. Points represent point estimates for the IRR with accompanying 95% confidence intervals. IRRs were derived using the ratio of predicted cases with exposure and predicted cases without exposure. Statistical significance was denoted with orange points when 95% CIs of the IRRs do not cross 1. (JPEG)

**S14 Fig. Incidence rate ratio of leptospirosis over days of SPI for each province.** Fig shows incidence rate ratio (IRR) of each disease over days of SPI for each province. The IRR gives the ratio of predicted cases with exposure to predicted cases with no exposure obtained from the disease-specific generalised additive models. Specifically, the IRR for SPI gives the ratio between incidence rates at varying values of SPI and the incidence rate at 0 SPI. An SPI value of 0 represents average precipitation conditions based on the long-term mean precipitation in that location. An IRR above 1 represents increased incidence rate with exposure compared to no exposure, while an IRR below 1 represents decreased incidence rate. Points represent point estimates for the IRR with accompanying 95% confidence intervals. IRRs were derived using the ratio of predicted cases with exposure and predicted cases without exposure. Statistical significance was denoted with orange points when 95% CIs of the IRRs do not cross 1.
(JPEG)

**S15 Fig. Incidence rate ratio of malaria over days of SPI for each province.** Fig shows incidence rate ratio (IRR) of each disease over days of SPI for each province. The IRR gives the ratio of predicted cases with exposure to predicted cases with no exposure obtained from the disease-specific generalised additive models. Specifically, the IRR for SPI gives the ratio between incidence rates at varying values of SPI and the incidence rate at 0 SPI. An SPI value of 0 represents average precipitation conditions based on the long-term mean precipitation in that location. An IRR above 1 represents increased incidence rate with exposure compared to no exposure, while an IRR below 1 represents decreased incidence rate. Points represent point estimates for the IRR with accompanying 95% confidence intervals. IRRs were derived using the ratio of predicted cases with exposure and predicted cases without exposure. Statistical significance was denoted with orange points when 95% CIs of the IRRs do not cross 1.
(JPEG)

**S16 Fig. Incidence rate ratio of melioidosis over days of SPI for each province.** Fig shows incidence rate ratio (IRR) of each disease over days of SPI for each province. The IRR gives the ratio of predicted cases with exposure to predicted cases with no exposure obtained from the disease-specific generalised additive models. Specifically, the IRR for SPI gives the ratio between incidence rates at varying values of SPI and the incidence rate at 0 SPI. An SPI value of 0 represents average precipitation conditions based on the long-term mean precipitation in that location. An IRR above 1 represents increased incidence rate with exposure compared to no exposure, while an IRR below 1 represents decreased incidence rate. Points represent point estimates for the IRR with accompanying 95% confidence intervals. IRRs were derived using the ratio of predicted cases with exposure and predicted cases without exposure. Statistical significance was denoted with orange points when 95% CIs of the IRRs do not cross 1.
(JPEG)

**S17 Fig. Incidence rate ratio of pneumonia over days of SPI for each province.** Fig shows incidence rate ratio (IRR) of each disease over days of SPI for each province. The IRR gives the ratio of predicted cases with exposure to predicted cases with no exposure obtained from the disease-specific generalised additive models. Specifically, the IRR for SPI gives the ratio between incidence rates at varying values of SPI and the incidence rate at 0 SPI. An SPI value of 0 represents average precipitation conditions based on the long-term mean precipitation in that location. An IRR above 1 represents increased incidence rate with exposure compared to no exposure, while an IRR below 1 represents decreased incidence rate. Points represent point estimates for the IRR with accompanying 95% confidence intervals. IRRs were derived using the ratio of predicted cases with exposure and predicted cases without exposure. Statistical significance was denoted with orange points when 95% CIs of the IRRs do not cross 1.
(JPEG)

**S18 Fig. Population attributable fraction of dengue over days of SPI and extreme heat for each province.** The population attributable fraction (PAF) gives the proportion of cases in a population that is attributed to a particular risk factor. A

PAF of 0 indicates that the risk factor has no impact on the occurrence of the disease, and a negative PAF indicates that the factor has a protective effect against the disease. The province-specific PAF was estimated with the ratio of predicted cases when varying either SPI or extreme heat days from zero, with other covariates held constant, against predicted cases when SPI and extreme heat days are set to zero. Predicted cases were obtained from disease-specific generalised additive models.

(PNG)

**S19 Fig. Population attributable fraction of Japanese encephalitis virus over days of SPI and extreme heat for each province.** The population attributable fraction (PAF) gives the proportion of cases in a population that is attributed to a particular risk factor. A PAF of 0 indicates that the risk factor has no impact on the occurrence of the disease, and a negative PAF indicates that the factor has a protective effect against the disease. The province-specific PAF was estimated with the ratio of predicted cases when varying either SPI or extreme heat days from zero, with other covariates held constant, against predicted cases when SPI and extreme heat days are set to zero. Predicted cases were obtained from disease-specific generalised additive models.

(PNG)

**S20 Fig. Population attributable fraction of influenza over days of SPI and extreme heat for each province.** The population attributable fraction (PAF) gives the proportion of cases in a population that is attributed to a particular risk factor. A PAF of 0 indicates that the risk factor has no impact on the occurrence of the disease, and a negative PAF indicates that the factor has a protective effect against the disease. The province-specific PAF was estimated with the ratio of predicted cases when varying either SPI or extreme heat days from zero, with other covariates held constant, against predicted cases when SPI and extreme heat days are set to zero. Predicted cases were obtained from disease-specific generalised additive models.

(PNG)

**S21 Fig. Population attributable fraction of leptospirosis over days of SPI and extreme heat for each province.** The population attributable fraction (PAF) gives the proportion of cases in a population that is attributed to a particular risk factor. A PAF of 0 indicates that the risk factor has no impact on the occurrence of the disease, and a negative PAF indicates that the factor has a protective effect against the disease. The province-specific PAF was estimated with the ratio of predicted cases when varying either SPI or extreme heat days from zero, with other covariates held constant, against predicted cases when SPI and extreme heat days are set to zero. Predicted cases were obtained from disease-specific generalised additive models.

(PNG)

**S22 Fig. Population attributable fraction of malaria over days of SPI and extreme heat for each province.** The population attributable fraction (PAF) gives the proportion of cases in a population that is attributed to a particular risk factor. A PAF of 0 indicates that the risk factor has no impact on the occurrence of the disease, and a negative PAF indicates that the factor has a protective effect against the disease. The province-specific PAF was estimated with the ratio of predicted cases when varying either SPI or extreme heat days from zero, with other covariates held constant, against predicted cases when SPI and extreme heat days are set to zero. Predicted cases were obtained from disease-specific generalised additive models.

(PNG)

**S23 Fig. Population attributable fraction of melioidosis over days of SPI and extreme heat for each province.** The population attributable fraction (PAF) gives the proportion of cases in a population that is attributed to a particular risk factor. A PAF of 0 indicates that the risk factor has no impact on the occurrence of the disease, and a negative PAF indicates

that the factor has a protective effect against the disease. The province-specific PAF was estimated with the ratio of predicted cases when varying either SPI or extreme heat days from zero, with other covariates held constant, against predicted cases when SPI and extreme heat days are set to zero. Predicted cases were obtained from disease-specific generalised additive models.
(PNG)

**S24 Fig. Population attributable fraction of pneumonia over days of SPI and extreme heat for each province.** The population attributable fraction (PAF) gives the proportion of cases in a population that is attributed to a particular risk factor. A PAF of 0 indicates that the risk factor has no impact on the occurrence of the disease, and a negative PAF indicates that the factor has a protective effect against the disease. The province-specific PAF was estimated with the ratio of predicted cases when varying either SPI or extreme heat days from zero, with other covariates held constant, against predicted cases when SPI and extreme heat days are set to zero. Predicted cases were obtained from disease-specific generalised additive models.
(PNG)

**S25 Fig. Lag–response curves showing the relative risk of seven infectious diseases (dengue, JEV, influenza, pneumonia, leptospirosis, melioidosis, and malaria) per 1-unit increase in standardized precipitation index (SPI) and extreme heat.** Relative risk (RR) represents the multiplicative change in disease incidence associated with a 1-unit increase in SPI or extreme heat at each lag compared to when all covariates are held at the mean.
(PNG)

**S26 Fig. Projected excess risk of diseases in extreme heat when relative humidity is held constant.** Excess risk associated with extreme heat under scenarios holding relative humidity constant (RH) (left panels) and using projected RH (right panels) for seven infectious diseases. Estimates are shown across climate change scenarios and time periods. Excess risk represents the percentage change in disease cases compared to historical levels. Holding RH constant results in minimal and non-significant dengue excess risk during extreme heat, suggesting that RH drives projected changes in dengue risk during periods of extreme heat.
(PNG)

**S27 Fig. Incidence rate ratio of diseases across different levels of relative humidity.** Fig shows incidence rate ratio (IRR) of each disease over relative humidity (RH). The IRR gives the ratio of predicted cases with the respective RH to predicted cases with mean RH of 75%, obtained from the disease-specific generalised additive models. An IRR above 1 represents increased incidence rate when relative humidity exceeds the mean, while an IRR below 1 represents decreased incidence rate. Points represent point estimates for the IRR with accompanying 95% confidence intervals. IRRs were derived using the ratio of predicted cases with exposure and predicted cases with mean exposure. Statistical significance was denoted with orange points when 95% CIs of the IRRs do not cross 1.
(PNG)

**S28 Fig. Province-level excess risk of dengue attributable to extreme weather across time and climate change scenarios.** Disease-specific generalised additive models were trained using historical data and future disease cases were projected based on MIROC6 general circulation model climate data. Province-level excess risk was calculated using the mean disease case counts across the historical period and the projected case counts at a respective time period and climate change scenario at each province. Excess risk represents the percentage change in disease cases compared to historical levels. Missing polygons are provinces which are not projected to experience extreme weather events in that period and climate change scenario. Asterisks represent provinces with statistically significant excess risk. Map created using GADM data (https://gadm.org/index.html, freely available for academic use). The map outlines and administrative boundaries are used with permission for academic publishing.
(PNG)

**S29 Fig. Province-level excess risk of Japanese Encephalitis attributable to extreme weather across time and climate change scenarios.** Disease-specific generalised additive models were trained using historical data and future disease cases were projected based on MIROC6 general circulation model climate data. Province-level excess risk was calculated using the mean disease case counts across the historical period and the projected case counts at a respective time period and climate change scenario at each province. Excess risk represents the percentage change in disease cases compared to historical levels. Missing polygons are provinces which are not projected to experience extreme weather events in that period and climate change scenario. Asterisks represent provinces with statistically significant excess risk. Map created using GADM data (https://gadm.org/index.html, freely available for academic use). The map outlines and administrative boundaries are used with permission for academic publishing.
(PNG)

**S30 Fig. Province-level excess risk of Japanese Encephalitis attributable to extreme weather across time and climate change scenarios.** Disease-specific generalised additive models were trained using historical data and future disease cases were projected based on MIROC6 general circulation model climate data. Province-level excess risk was calculated using the mean disease case counts across the historical period and the projected case counts at a respective time period and climate change scenario at each province. Excess risk represents the percentage change in disease cases compared to historical levels. Missing polygons are provinces which are not projected to experience extreme weather events in that period and climate change scenario. Asterisks represent provinces with statistically significant excess risk. Map created using GADM data (https://gadm.org/index.html, freely available for academic use). The map outlines and administrative boundaries are used with permission for academic publishing.
(PNG)

**S31 Fig. Province-level excess risk of leptospirosis attributable to extreme weather across time and climate change scenarios.** Disease-specific generalised additive models were trained using historical data and future disease cases were projected based on MIROC6 general circulation model climate data. Province-level excess risk was calculated using the mean disease case counts across the historical period and the projected case counts at a respective time period and climate change scenario at each province. Excess risk represents the percentage change in disease cases compared to historical levels. Missing polygons are provinces which are not projected to experience extreme weather events in that period and climate change scenario. Asterisks represent provinces with statistically significant excess risk. Map created using GADM data (https://gadm.org/index.html, freely available for academic use). The map outlines and administrative boundaries are used with permission for academic publishing.
(PNG)

**S32 Fig. Province-level excess risk of malaria attributable to extreme weather across time and climate change scenarios.** Disease-specific generalised additive models were trained using historical data and future disease cases were projected based on MIROC6 general circulation model climate data. Province-level excess risk was calculated using the mean disease case counts across the historical period and the projected case counts at a respective time period and climate change scenario at each province. Excess risk represents the percentage change in disease cases compared to historical levels. Missing polygons are provinces which are not projected to experience extreme weather events in that period and climate change scenario. Asterisks represent provinces with statistically significant excess risk. Map created using GADM data (https://gadm.org/index.html, freely available for academic use). The map outlines and administrative boundaries are used with permission for academic publishing.
(PNG)

**S33 Fig. Province-level excess risk of melioidosis attributable to extreme weather across time and climate change scenarios.** Disease-specific generalised additive models were trained using historical data and future disease

cases were projected based on MIROC6 general circulation model climate data. Province-level excess risk was calculated using the mean disease case counts across the historical period and the projected case counts at a respective time period and climate change scenario at each province. Excess risk represents the percentage change in disease cases compared to historical levels. Missing polygons are provinces which are not projected to experience extreme weather events in that period and climate change scenario. Asterisks represent provinces with statistically significant excess risk. Map created using GADM data (https://gadm.org/index.html, freely available for academic use). The map outlines and administrative boundaries are used with permission for academic publishing.
(PNG)

**S34 Fig. Province-level excess risk of pneumonia attributable to extreme weather across time and climate change scenarios.** Disease-specific generalised additive models were trained using historical data and future disease cases were projected based on MIROC6 general circulation model climate data. Province-level excess risk was calculated using the mean disease case counts across the historical period and the projected case counts at a respective time period and climate change scenario at each province. Excess risk represents the percentage change in disease cases compared to historical levels. Missing polygons are provinces which are not projected to experience extreme weather events in that period and climate change scenario. Asterisks represent provinces with statistically significant excess risk. Map created using GADM data (https://gadm.org/index.html, freely available for academic use). The map outlines and administrative boundaries are used with permission for academic publishing.
(PNG)

**S35 Fig. Province-level distribution of extreme heat and excess risks of dengue under SSP245.** (A) Average number of extreme heat days in a month by province from 2021-2100. (B) Province-level excess risks of dengue under SSP245 from 2021–2100. Asterisks indicate a statistically significant excess risk value. Excess risk represents the percentage change in annual disease case counts from the historical baseline from 2003 to 2020. Map created using GADM data (https://gadm.org/index.html, freely available for academic use). The map outlines and administrative boundaries are used with permission for academic publishing.
(PNG)

**S36 Fig. Province-level distribution of extreme heat and excess risks of influenza under SSP245.** (A) Average number of extreme heat days in a month by province from 2021-2100. (B) Province-level excess risks of dengue under SSP245 from 2021–2100. Asterisks indicate a statistically significant excess risk value. Excess risk represents the percentage change in annual disease case counts from the historical baseline from 2003 to 2020. Map created using GADM data (https://gadm.org/index.html, freely available for academic use). The map outlines and administrative boundaries are used with permission for academic publishing.
(PNG)

**S37 Fig. Excess risk of dengue attributable to extreme weather taking into account future population.** Annual excess risk of dengue in Thailand attributed to (A) all extreme weather events, (B) extreme heat days, (C) extreme dry weather and (D) extreme wet weather. Disease-specific generalized additive models (GAM) were trained with lagged extreme heat days, lagged standardized precipitation index (SPI), relative humidity and population density as variables, and population was used as an offset. Future disease cases were then projected using future extreme heat days, SPI, relative humidity from the MIROC6 general circulation model but historical population density, and historical population as an offset. In a separate analysis, the GAMs were trained on a separate historical population dataset which corresponds to the future population dataset, and we projected disease cases using future population density as a variable and future population as an offset. National-level excess risk was calculated using the mean disease case counts across the historical period, 2003–2019, and the projected case counts at a respective time period and climate change scenario at the national

level. Excess risk represents the percentage change in disease cases compared to historical levels. National-level excess risks of each disease were observed to take very extreme values and hence, we did not use future population and population density when projecting disease cases.
(PNG)

**S38 Fig. Excess risk of Japanese encephalitis attributable to extreme weather taking into account future population.** Annual excess risk of JEV in Thailand attributed to (A) all extreme weather events, (B) extreme heat days, (C) extreme dry weather and (D) extreme wet weather. Disease-specific generalized additive models (GAM) were trained with lagged extreme heat days, lagged standardized precipitation index (SPI), relative humidity and population density as variables, and population was used as an offset. Future disease cases were then projected using future extreme heat days, SPI, relative humidity from the MIROC6 general circulation model but historical population density, and historical population as an offset. In a separate analysis, the GAMs were trained on a separate historical population dataset which corresponds to the future population dataset, and we projected disease cases using future population density as a variable and future population as an offset. National-level excess risk was calculated using the mean disease case counts across the historical period, 2003–2019, and the projected case counts at a respective time period and climate change scenario at the national level. Excess risk represents the percentage change in disease cases compared to historical levels. National-level excess risks of each disease were observed to take very extreme values and hence, we did not use future population and population density when projecting disease cases.
(PNG)

**S39 Fig. Excess risk of influenza attributable to extreme weather taking into account future population.** Annual excess risk of influenza in Thailand attributed to (A) all extreme weather events, (B) extreme heat days, (C) extreme dry weather and (D) extreme wet weather. Missing values are due to insufficient data in that period. Disease-specific generalized additive models (GAM) were trained with lagged extreme heat days, lagged standardized precipitation index (SPI), relative humidity and population density as variables, and population was used as an offset. Future disease cases were then projected using future extreme heat days, SPI, relative humidity from the MIROC6 general circulation model but historical population density, and historical population as an offset. In a separate analysis, the GAMs were trained on a separate historical population dataset which corresponds to the future population dataset, and we projected disease cases using future population density as a variable and future population as an offset. National-level excess risk was calculated using the mean disease case counts across the historical period, 2003–2019, and the projected case counts at a respective time period and climate change scenario at the national level. Excess risk represents the percentage change in disease cases compared to historical levels. National-level excess risks of each disease were observed to take very extreme values and hence, we did not use future population and population density when projecting disease cases.
(PNG)

**S40 Fig. Excess risk of leptospirosis attributable to extreme weather taking into account future population.** Annual excess risk of leptospirosis in Thailand attributed to (A) all extreme weather events, (B) extreme heat days, (C) extreme dry weather and (D) extreme wet weather. Disease-specific generalized additive models (GAM) were trained with lagged extreme heat days, lagged standardized precipitation index (SPI), relative humidity and population density as variables, and population was used as an offset. Future disease cases were then projected using future extreme heat days, SPI, relative humidity from the MIROC6 general circulation model but historical population density, and historical population as an offset. In a separate analysis, the GAMs were trained on a separate historical population dataset which corresponds to the future population dataset, and we projected disease cases using future population density as a variable and future population as an offset. National-level excess risk was calculated using the mean disease case counts across the historical period, 2003–2019, and the projected case counts at a respective time period and climate change scenario at the national level. Excess risk represents the percentage change in disease cases compared to historical

levels. National-level excess risks of each disease were observed to take very extreme values and hence, we did not use future population and population density when projecting disease cases.
(PNG)

**S41 Fig. Excess risk of malaria attributable to extreme weather taking into account future population.** Annual excess risk of malaria in Thailand attributed to (A) all extreme weather events, (B) extreme heat days, (C) extreme dry weather and (D) extreme wet weather. Disease-specific generalized additive models (GAM) were trained with lagged extreme heat days, lagged standardized precipitation index (SPI), relative humidity and population density as variables, and population was used as an offset. Future disease cases were then projected using future extreme heat days, SPI, relative humidity from the MIROC6 general circulation model but historical population density, and historical population as an offset. In a separate analysis, the GAMs were trained on a separate historical population dataset which corresponds to the future population dataset, and we projected disease cases using future population density as a variable and future population as an offset. National-level excess risk was calculated using the mean disease case counts across the historical period, 2003–2019, and the projected case counts at a respective time period and climate change scenario at the national level. Excess risk represents the percentage change in disease cases compared to historical levels. National-level excess risks of each disease were observed to take very extreme values and hence, we did not use future population and population density when projecting disease cases.
(PNG)

**S42 Fig. Excess risk of melioidosis attributable to extreme weather taking into account future population.** Annual excess risk of melioidosis in Thailand attributed to (A) all extreme weather events, (B) extreme heat days, (C) extreme dry weather and (D) extreme wet weather. Disease-specific generalized additive models (GAM) were trained with lagged extreme heat days, lagged standardized precipitation index (SPI), relative humidity and population density as variables, and population was used as an offset. Future disease cases were then projected using future extreme heat days, SPI, relative humidity from the MIROC6 general circulation model but historical population density, and historical population as an offset. In a separate analysis, the GAMs were trained on a separate historical population dataset which corresponds to the future population dataset, and we projected disease cases using future population density as a variable and future population as an offset. National-level excess risk was calculated using the mean disease case counts across the historical period, 2003–2019, and the projected case counts at a respective time period and climate change scenario at the national level. Excess risk represents the percentage change in disease cases compared to historical levels. National-level excess risks of each disease were observed to take very extreme values and hence, we did not use future population and population density when projecting disease cases.
(PNG)

**S43 Fig. Excess risk of pneumonia attributable to extreme weather taking into account future population.** Annual excess risk of pneumonia in Thailand attributed to (A) all extreme weather events, (B) extreme heat days, (C) extreme dry weather and (D) extreme wet weather. Disease-specific generalized additive models (GAM) were trained with lagged extreme heat days, lagged standardized precipitation index (SPI), relative humidity and population density as variables, and population was used as an offset. Future disease cases were then projected using future extreme heat days, SPI, relative humidity from the MIROC6 general circulation model but historical population density, and historical population as an offset. In a separate analysis, the GAMs were trained on a separate historical population dataset which corresponds to the future population dataset, and we projected disease cases using future population density as a variable and future population as an offset. National-level excess risk was calculated using the mean disease case counts across the historical period, 2003–2019, and the projected case counts at a respective time period and climate change scenario at the national level. Excess risk represents the percentage change in disease cases compared to historical levels. National-level excess

risks of each disease were observed to take very extreme values and hence, we did not use future population and population-density when projecting disease cases.
(PNG)

**S44 Fig. Excess risk of pneumonia, influenza, JEV, malaria, dengue, melioidosis and leptospirosis during periods of extreme weather, across 2021–2100, in climate change scenarios SSP126, SSP245, SSP370 and SSP585 using CMCC-ESM2 climate change model.** Disease-specific generalized additive models (GAM) were trained on historical data from 2003-2019 and used to project future case counts of the respective disease across 4 time periods (2021–2040, 2041–2060, 2061–2080, 2081–2100) and 4 climate change scenarios (SSP126, SSP245, SSP370, SSP585) during periods of extreme weather. Projections were made based on climate data from the CMCC-ESM2 general circulation model. National-level excess risk was calculated using the mean disease case counts across the historical period and the projected case counts at a respective time period and climate change scenario. Excess risk represents the percentage change in disease cases compared to historical levels.
(PNG)

**S45 Fig. Excess risk of pneumonia, influenza, JEV, malaria, dengue, melioidosis and leptospirosis during periods of extreme weather, across 2021–2100, in climate change scenarios SSP126, SSP245, SSP370 and SSP585 using IPSL-CM6A-LR climate change model.** Disease-specific generalized additive models (GAM) were trained on historical data from 2003-2019 and used to project future case counts of the respective disease across 4 time periods (2021–2040, 2041–2060, 2061–2080, 2081–2100) and 4 climate change scenarios (SSP126, SSP245, SSP370, SSP585) during periods of extreme weather. Projections were made based on climate data from the IPSL-CM6A-LR general circulation model. National-level excess risk was calculated using the mean disease case counts across the historical period and the projected case counts at a respective time period and climate change scenario. Excess risk represents the percentage change in disease cases compared to historical levels.
(PNG)

**S46 Fig. Excess risk of pneumonia, influenza, JEV, malaria, dengue, melioidosis and leptospirosis during periods of extreme weather, across 2021–2100, in climate change scenarios SSP126, SSP245, SSP370 and SSP585 using multi-model ensemble climate change model.** Disease-specific generalized additive models (GAM) were trained on historical data from 2003-2019 and used to project future case counts of the respective disease across 4 time periods (2021–2040, 2041–2060, 2061–2080, 2081–2100) and 4 climate change scenarios (SSP126, SSP245, SSP370, SSP585) during periods of extreme weather. Projections were made based on the means of climate data from the MIROC6, CMCC-ESM2 and IPSL-CM6A-LR general circulation model. National-level excess risk was calculated using the mean disease case counts across the historical period and the projected case counts at a respective time period and climate change scenario. Excess risk represents the percentage change in disease cases compared to historical levels.
(PNG)

**S47 Fig. Excess risk (ER) of JEV, malaria and dengue during periods of extreme heat, extreme dry weather and extreme wet weather, across 2021–2100, in climate change scenarios SSP126, SSP245, SSP370 and SSP585 using CMCC-ESM2 climate change model.** Disease-specific generalized additive models (GAM) were trained on historical data from 2003-2019 and used to project future case counts of the respective disease across 4 time periods (2021–2040, 2041–2060, 2061–2080, 2081–2100) and 4 climate change scenarios (SSP126, SSP245, SSP370, SSP585) during periods of extreme weather. Projections were made based on climate data from the CMCC-ESM2 general circulation model. National-level excess risk was calculated using the mean disease case counts across the historical period and the projected case counts at a respective time period and climate change scenario. Excess risk was stratified across each type of extreme weather, which represents the percentage change in disease cases compared to historical levels during

periods of extreme heat, extreme dry weather and extreme wet weather alone. The results obtained using the CMCC-ESM2 general circulation model were generally similar to the main results obtained using the MIROC6 general circulation model.
(PNG)

**S48 Fig. Excess risk (ER) of JEV, malaria and dengue during periods of extreme heat, extreme dry weather and extreme wet weather, across 2021–2100, in climate change scenarios SSP126, SSP245, SSP370 and SSP585 using IPSL-CM6A-LR climate change model.** Disease-specific generalized additive models (GAM) were trained on historical data from 2003-2019 and used to project future case counts of the respective disease across 4 time periods (2021–2040, 2041–2060, 2061–2080, 2081–2100) and 4 climate change scenarios (SSP126, SSP245, SSP370, SSP585) during periods of extreme weather. Projections were made based on climate data from the IPSL-CM6A-LR general circulation model. National-level excess risk was calculated using the mean disease case counts across the historical period and the projected case counts at a respective time period and climate change scenario. Excess risk was stratified across each type of extreme weather, which represents the percentage change in disease cases compared to historical levels during periods of extreme heat, extreme dry weather and extreme wet weather alone. The results obtained using the CMCC-ESM2 general circulation model were slightly different from the main results obtained using the MIROC6 general circulation model. Dengue risk is expected to increase during periods of extreme dry weather in all climate change scenarios, which varies from the main model where dengue risk is expected to decrease during periods of extreme dry weather.
(PNG)

**S49 Fig. Excess risk (ER) of influenza, pneumonia, leptospirosis and melioidosis during periods of extreme heat, extreme dry weather and extreme wet weather, across 2021–2100, in climate change scenarios SSP126, SSP245, SSP370 and SSP585 using CMCC-ESM2 climate change model.** Disease-specific generalized additive models (GAM) were trained on historical data from 2003-2019 and used to project future case counts of the respective disease across 4 time periods (2021–2040, 2041–2060, 2061–2080, 2081–2100) and 4 climate change scenarios (SSP126, SSP245, SSP370, SSP585) during periods of extreme weather. Projections were made based on climate data from the CMCC-ESM2 general circulation model. National-level excess risk was calculated using the mean disease case counts across the historical period and the projected case counts at a respective time period and climate change scenario. Excess risk was stratified across each type of extreme weather, which represents the percentage change in disease cases compared to historical levels during periods of extreme heat, extreme dry weather and extreme wet weather alone. The results obtained using the CMCC-ESM2 general circulation model were generally similar to the main results obtained using the MIROC6 general circulation model.
(PNG)

**S50 Fig. Excess risk (ER) of influenza, pneumonia, leptospirosis and melioidosis during periods of extreme heat, extreme dry weather and extreme wet weather, across 2021–2100, in climate change scenarios SSP126, SSP245, SSP370 and SSP585 using IPSL-CM6A-LR climate change model.** Disease-specific generalized additive models (GAM) were trained on historical data from 2003-2019 and used to project future case counts of the respective disease across 4 time periods (2021–2040, 2041–2060, 2061–2080, 2081–2100) and 4 climate change scenarios (SSP126, SSP245, SSP370, SSP585) during periods of extreme weather. Projections were made based on climate data from the IPSL-CM6A-LR general circulation model. National-level excess risk was calculated using the mean disease case counts across the historical period and the projected case counts at a respective time period and climate change scenario. Excess risk was stratified across each type of extreme weather, which represents the percentage change in disease cases compared to historical levels during periods of extreme heat, extreme dry weather and extreme wet weather alone. The results obtained using the CMCC-ESM2 general circulation model were largely similar to the main results obtained

using the MIROC6 general circulation model. The expected increase in influenza during periods of extreme wet and dry weather are much higher than projected in the main model.
(PNG)

**S51 Fig. Excess risk of diseases attributable to extreme heat sub-scenarios.** Disease-specific generalized additive models (GAM) were trained on historical data from 2003-2019 and used to project future case counts of the respective disease across 4 time periods (2021–2040, 2041–2060, 2061–2080, 2081–2100) and 4 climate change scenarios (SSP126, SSP245, SSP370, SSP585) during periods of extreme weather. National-level excess risk was calculated using the mean disease case counts across the historical period and the projected case counts at a respective time period and climate change scenario. Excess risk represents the percentage change in disease cases compared to historical levels. We estimated the excess risk attributable to extreme heat, controlling for the presence or absence of concurrent extreme wet weather or extreme dry weather. Excess risks attributable to extreme heat resembled the excess risks attributable to extreme heat with normal dry/wet weather. This indicates that the excess risks attributable to extreme heat were not confounded by the effects of extreme dry/wet weather.
(PNG)

**S52 Fig. Excess risk of diseases attributable to extreme dry weather sub-scenarios.** Disease-specific generalized additive models (GAM) were trained on historical data from 2003-2019 and used to project future case counts of the respective disease across 4 time periods (2021–2040, 2041–2060, 2061–2080, 2081–2100) and 4 climate change scenarios (SSP126, SSP245, SSP370, SSP585) during periods of extreme weather. National-level excess risk was calculated using the mean disease case counts across the historical period and the projected case counts at a respective time period and climate change scenario. Excess risk represents the percentage change in disease cases compared to historical levels. We estimated the excess risk attributable to extreme dry weather, controlling for the presence or absence of concurrent extreme heat. Excess risks attributable to extreme dry weather resembled the excess risks attributable to extreme dry weather with no extreme heat. This indicates that the excess risks attributable to extreme dry weather were not confounded by the effects of extreme heat.
(PNG)

**S53 Fig. Excess risk of diseases attributable to extreme wet weather sub-scenarios.** Disease-specific generalized additive models (GAM) were trained on historical data from 2003-2019 and used to project future case counts of the respective disease across 4 time periods (2021–2040, 2041–2060, 2061–2080, 2081–2100) and 4 climate change scenarios (SSP126, SSP245, SSP370, SSP585) during periods of extreme weather. National-level excess risk was calculated using the mean disease case counts across the historical period and the projected case counts at a respective time period and climate change scenario. Excess risk represents the percentage change in disease cases compared to historical levels. We estimated the excess risk attributable to extreme wet weather, controlling for the presence or absence of concurrent extreme heat. The excess risk attributable to extreme wet weather paralleled the excess risks attributable to a combination of extreme wet weather and no extreme heat. This suggests that the excess risks attributable to extreme wet weather were similarly not confounded by the effects of extreme heat.
(PNG)

## Author contributions

**Data curation:** Esther Li Wen Choo, Pei Ma, Jo Yi Chow.

**Formal analysis:** Esther Li Wen Choo.

**Funding acquisition:** Jue Tao Lim.

**Investigation:** Esther Li Wen Choo, Pei Ma, Jo Yi Chow, Jue Tao Lim.

**Methodology:** Esther Li Wen Choo, Steve Hung-Lam Yim, Oliver Brady, Borame Lee Dickens, Jue Tao Lim.

**Project administration:** Jue Tao Lim.

**Resources:** Jue Tao Lim.

**Software:** Esther Li Wen Choo, Pei Ma.

**Supervision:** Jue Tao Lim.

**Validation:** Esther Li Wen Choo.

**Visualization:** Esther Li Wen Choo.

**Writing – original draft:** Esther Li Wen Choo, Jue Tao Lim.

**Writing – review & editing:** Esther Li Wen Choo, Steve Hung-Lam Yim, Oliver Brady, Borame Lee Dickens, Jue Tao Lim.

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
