## [Decision Letter · Decision Letter 0]

10 Oct 2025

Projecting long-term excess risks of major infectious diseases associated with future extreme weather events in Thailand

Dear Dr. Choo,

Thank you for submitting your manuscript to PLOS Neglected Tropical Diseases. After careful consideration, we feel that it has merit but does not fully meet PLOS Neglected Tropical Diseases's publication criteria as it currently stands. Therefore, we invite you to submit a revised version of the manuscript that addresses the points raised during the review process.

Please submit your revised manuscript within 60 days Dec 09 2025 11:59PM. If you will need more time than this to complete your revisions, please reply to this message or contact the journal office at plosntds@plos.org. Please include the following items when submitting your revised manuscript:

We look forward to receiving your revised manuscript.

Kind regards,

Marilia Sá Carvalho

Academic Editor

Qu Cheng

Section Editor

Shaden Kamhawi

co-Editor-in-Chief

Paul Brindley

co-Editor-in-Chief

**Journal Requirements:**

At this stage, the following Authors/Authors require contributions: Esther Li Wen Choo, Pei Ma, Jo Yi Chow, Steve Hung-Lam Yim, Oliver Brady, Borame Lee Dickens, and Jue Tao Lim. Please ensure that the full contributions of each author are acknowledged in the "Add/Edit/Remove Authors" section of our submission form.

Potential Copyright Issues:

i) Figures 6, 7, S1-S3, and S25-S31. Please (a) provide a direct link to the base layer of the map (i.e., the country or region border shape) and ensure this is also included in the figure legend; and (b) provide a link to the terms of use / license information for the base layer image or shapefile. We cannot publish proprietary or copyrighted maps (e.g. Google Maps, Mapquest) and the terms of use for your map base layer must be compatible with our CC BY 4.0 license.

6) Kindly revise your competing statement in the online submission form to align with the journal's style guidelines: 'The authors declare that there are no competing interests.'

**Reviewers' Comments:**

Reviewer's Responses to Questions

**Key Review Criteria Required for Acceptance?**

**Methods**

-Are the objectives of the study clearly articulated with a clear testable hypothesis stated?

-Is the study design appropriate to address the stated objectives?

-Is the population clearly described and appropriate for the hypothesis being tested?

-Is the sample size sufficient to ensure adequate power to address the hypothesis being tested?

-Were correct statistical analysis used to support conclusions?

-Are there concerns about ethical or regulatory requirements being met?

Reviewer #1: In general, the methods seem to be sound. In the attached comments I document some queries about equation 1 labeling as well as how lags were treated. I was also curious about the use of extreme heat rather than temperature.

Reviewer #2: (No Response)

**Results**

-Does the analysis presented match the analysis plan?

-Are the results clearly and completely presented?

-Are the figures (Tables, Images) of sufficient quality for clarity?

Reviewer #1: See comments - there were several places where I felt the results did not match with Figure 2. I was confused about the apparent mismatch between some IRRs and some PAFs. The reference for SPI did not seem to be 0 as expected. Figure 3 was extremely busy, as was Figure 5A and 6A.

Reviewer #2: (No Response)

**Conclusions**

-Are the conclusions supported by the data presented?

-Are the limitations of analysis clearly described?

-Do the authors discuss how these data can be helpful to advance our understanding of the topic under study?

-Is public health relevance addressed?

Reviewer #1: (No Response)

Reviewer #2: (No Response)

**Editorial and Data Presentation Modifications?**

Reviewer #1: (No Response)

Reviewer #2: (No Response)

**Summary and General Comments**

Reviewer #1: (No Response)

Reviewer #2: Major comments

• Extreme heat threshold: You set the national 90th percentile Tmax (34.9 °C) and tally the number of days in a month that have at least three days of heat. This might put heat in the wrong category in milder northern and highland areas and downplay heat in hotter southern and coastal provinces. Please test percentiles that are particular to a province or climatic zone and let us know how sensitive IRR/ER is to that decision.

• You utilise SPI-1 (no PET) since PET projections are restricted. Think about utilising SPEI-1 with PET approximations (such Hargreaves or Penman-Monteith based on CMIP6 variables) to see how sensitive it is, or at least prove that SPI-1 and SPEI-1 categorise dry and wet months in the same way for a historical overlap. This will make it easier to draw conclusions about how heat and moisture interact.

• MODAWEC and linearly interpolated wet-day counts are used to create daily sequences for heatwaves and wet days from monthly averages. This may smooth out persistence and tails, which might change the durations of intense heat and heavy rain periods that affect transmission and exposure paths. Please compare the outputs of the generators to the reanalysis (ERA5) for the years 2003 to 2019 on metrics such the distribution of heatwave durations, the lengths of wet-day spells, and the daily precipitation at the 95th and 99th percentiles at the provincial level. Please quantify any attenuation and change the predictions or uncertainty as needed.

• Monthly case numbers very definitely show some degree of autocorrelation. The GAM framework has seasonality but not a clear AR term. Please report the remaining ACF/PACF and, if necessary, include AR(1) other distributed-lag factors (for example, corAR1) and re-fit the main models to make sure that the IRR/ER is stable.

• Province random intercepts provide for variability but fail to address spatial dependency. Examine a spatially structured effect (CAR/SAR) or spatial smooth of provincial centroids and see whether spatially informed models significantly alter IRR/ER.

• In addition to AIC vs. GLM, use rolling-origin time-series cross-validation (for example, 5-fold blocks) to present RMSE/MAE and the calibration of prediction intervals for 2015–2019. This will make the projections more believable.

• You set the population at historical averages to see how weather affects it; the other option is to use estimates that are slanted towards the future population. Instead of completely getting rid of demographics, please (i) show two sets of results: risk (per-capita incidence; your main) and burden (cases using SSP-consistent population trajectories) with clear caveats, and (ii) limit the effect by age-structure scenarios or a scaled offset that is capped at historical maxima to avoid extrapolation that goes too far. This helps policymakers understand the difference between instances avoided and cases that are too high compared to risk.

• You know that malaria and JEV rely on how much the forest is disturbed and how many pigs there are, which are both left out. Dengue also relies on how well water is managed in the developed environment. To cut down on omitted-variable bias, please add province-level time-varying proxies (such forest loss, pig census, urbanization/impervious surface, and WASH indexes) or at least provincial fixed trends. Give a directed acyclic graph (DAG) that shows the assumed routes.

• ER is calculated just during months designated as severe and is compared to historical baseline months. Because the incidence of exposure varies across SSPs, the selection of months is contingent upon the model used. Please include a counterfactual study by simulating each future month twice: once with forecast weather and once with "normal" weather based on historical monthly averages. Keep all other variables the same and report the difference. This sets up ER as a causal difference and prevents exposure from affecting it.

• You utilise three GCMs: MIROC6 for the main one, CMCC-ESM2 for the sensitivity one, and IPSL-CM6A-LR for the sensitivity one. Think about presenting multi-model ensemble ER medians and 5–95% ranges for a larger CMIP6 subset, or explain your choice by using model performance weights for monsoon indicators in Thailand. Also, make it clear whether the MIROC6 fields were modified for bias before MODAWEC.

• The abstract says that tuberculosis is one among the illnesses, yet the Methods section contains seven diseases (dengue, JEV, malaria, pneumonia, influenza, leptospirosis, and melioidosis) that do not include TB. Please make sure that all parts and figures match up with the same set.

• Under SSP245 (central/northern provinces), you see dengue cases rise from 2021 to 2060, then fall as temperatures rise and dry/wet extremes become more common. Influenza cases rise at first, but fall as circumstances become wetter. Please provide mechanistic explanation (for example, vector survival/EIP vs. adult mortality at high Tmax; human indoor mixing and absolute humidity for influenza) and exhibit ER maps next to anticipated heat-day distributions to visibly connect mechanism and location.

• Translate province-level ER into operational triggers (e.g., thresholds for extreme-heat days or SPI indicating ≥20% ER for dengue/influenza) and suggest specific actions (time for reducing vector sources, preparing hospitals for surges, and communicating risks) by area and decade. This will make the document more useful in real life.

Minor comments

• The table/methods only mention SSP470 once; in other places, you use SSP370. Please always fix things.

• You choose lags based on the average AIC of all the models for an illness. Briefly explain why the chosen delays (such dengue EIP/hatching windows and influenza incubation/transmission) make sense from a biological point of view, and provide an extra lag-response figure.

• You include RH at the same time. Think about measuring absolute humidity (which is more directly related to how easily the flu spreads) and lagged RH for strength; summarise in the Supplement.

• Make it clear whether monthly smooths are cyclical (January ≈ December). If not, think of a cyclic spline.

• When looking at IRR over a lot of SPI/heat values and provinces, keep in mind how you show uncertainty (emphasis on impact sizes and CIs instead of binary significance).

• Fig. 1: To indicate spread, add units (days, SPI unitless) and display distributions (like boxplots) for each time.

• Make sure the colour scales are the same and include the number of severe months that went into each ER estimate (this helps make sense of broad CIs).

• When you state "increase/decrease during periods of extreme weather," make it clear which event (hot, dry, or wet) you mean to prevent confusion, unless you're using the combined term.

• It's great that you provide a link to GitHub. Please pin the code to a release/tag, add an environment file, and put the processed province-level series in a data repository with a DOI.

• To make it easier to read, some sentences could be shortened (for example, there are two statements about flu rates going down in the second half of the century—merge and cross-reference the figure panels). • Briefly explain how you dealt with any gaps in case definition stability (for example, how you handled the province split/merge that was already mentioned).

• After fitting the GAMs, did you check for residual autocorrelation and spatial dependence? What did Moran's I and the residual ACF indicate if so?

• How much do your dengue/influenza ER findings change when you use provincial-specific heat thresholds instead of a nationwide threshold? For example, the 90th percentile per province.

• Did you fix any bias in the MIROC6/CMCC/IPSL inputs (for example, by using quantile mapping) before MODAWEC? If so, please write out how you did it; if not, please talk about what this means for extremes.

• How many intense months did each ER cell (disease × period × SSP) get? Could sparse counts be causing broad CIs, such in the case of melioidosis under SSP585?

• For each SSP, can you make a "normal weather" projection series that shows ER as a difference in anticipated incidence while keeping weather around historical means?

• Did model calibration or diagnostics take into account any surveillance artefacts, such COVID-19 interference around 2020 or 2009 H1N1 for influenza?

PLOS authors have the option to publish the peer review history of their article (what does this mean? ). If published, this will include your full peer review and any attached files.

**Do you want your identity to be public for this peer review?** For information about this choice, including consent withdrawal, please see our Privacy Policy .

Reviewer #1: No

Reviewer #2: No

**Figure resubmission:**
---

## [Decision Letter · Decision Letter 1]

7 Dec 2025

Response to Reviewers
Revised Manuscript with Track Changes
Manuscript

Shaden Kamhawi

co-Editor-in-Chief

Paul Brindley

co-Editor-in-Chief

**Journal Requirements:**

**Reviewers' comments:**

**Key Review Criteria Required for Acceptance?**

**Methods**

-Are the objectives of the study clearly articulated with a clear testable hypothesis stated?

-Is the study design appropriate to address the stated objectives?

-Is the population clearly described and appropriate for the hypothesis being tested?

-Is the sample size sufficient to ensure adequate power to address the hypothesis being tested?

-Were correct statistical analysis used to support conclusions?

-Are there concerns about ethical or regulatory requirements being met?

Reviewer #1: (No Response)

Reviewer #2: (No Response)

**Results**

-Does the analysis presented match the analysis plan?

-Are the results clearly and completely presented?

-Are the figures (Tables, Images) of sufficient quality for clarity?

Reviewer #1: (No Response)

Reviewer #2: (No Response)

**Conclusions**

-Are the conclusions supported by the data presented?

-Are the limitations of analysis clearly described?

-Do the authors discuss how these data can be helpful to advance our understanding of the topic under study?

-Is public health relevance addressed?

Reviewer #1: (No Response)

Reviewer #2: (No Response)

**Editorial and Data Presentation Modifications?**

Reviewer #1: I feel that there remains room for improvement in terms of simplifying the presentation of figures and results (especially Figs 3, 6, and 7).

Reviewer #2: (No Response)

**Summary and General Comments**

Reviewer #1: (No Response)

Reviewer #2: The authors have carefully addressed all reviewer comments and substantially improved the manuscript in terms of clarity, coherence, and methodological soundness. I am satisfied with the revisions and recommend the paper for publication in PLOS Neglected Tropical Diseases. Accept.

PLOS authors have the option to publish the peer review history of their article (what does this mean? ). If published, this will include your full peer review and any attached files.

**Do you want your identity to be public for this peer review?** For information about this choice, including consent withdrawal, please see our Privacy Policy .

Reviewer #1: No

Reviewer #2: No

**Figure resubmission:**

**Reproducibility:** To enhance the reproducibility of your results, we recommend that authors of applicable studies deposit laboratory protocols in protocols.io, where a protocol can be assigned its own identifier (DOI) such that it can be cited independently in the future. Additionally, PLOS ONE offers an option to publish peer-reviewed clinical study protocols. Read more information on sharing protocols at https://plos.org/protocols?utm_medium=editorial-email&utm_source=authorletters&utm_campaign=protocols

---

## [Editor Report · Decision Letter 2]

26 Dec 2025

Dear Ms Choo,

We are pleased to inform you that your manuscript 'Projecting long-term excess risks of major infectious diseases associated with future extreme weather events in Thailand' has been provisionally accepted for publication in PLOS Neglected Tropical Diseases.

Best regards,

Qu Cheng, Ph.D.

Section Editor

Qu Cheng

Section Editor

Shaden Kamhawi

co-Editor-in-Chief

Paul Brindley

co-Editor-in-Chief

---

## [Editor Report · Acceptance letter]

Dear Ms Choo,

We are delighted to inform you that your manuscript, " 

Projecting long-term excess risks of major infectious diseases associated with future extreme weather events in Thailand," has been formally accepted for publication in PLOS Neglected Tropical Diseases.

Best regards,

Shaden Kamhawi

co-Editor-in-Chief

Paul Brindley

co-Editor-in-Chief
